# CAN ADVERSARIAL SAMPLES BENEFIT FEW-SHOT UNSUPERVISED IMPLICIT NEURAL SHAPE REPRESENTATION LEARNING ?

## ABSTRACT

Implicit Neural Representations have gained prominence as a powerful framework for capturing complex data modalities, encompassing a wide range from 3D shapes to images and audio. Within the realm of 3D shape representation, Neural Signed Distance Functions (SDF) have demonstrated remarkable potential in faithfully encoding intricate shape geometry. However, learning SDFs from 3D point clouds in the absence of ground truth supervision remains a very challenging task. While recent methods rely on smoothness priors to regularize the learning, our method introduces a regularization term that leverages adversarial samples around the shape to improve the learned SDFs. Through extensive experiments and evaluations, we illustrate the efficacy of our proposed method, highlighting its capacity to improve SDF learning with respect to baselines and the state-of-the-art using synthetic and real data.

## 1 INTRODUCTION

Obtaining faith-full and intelligible neural representations of the 3D world from limited and corrupted point clouds is a challenge of paramount importance, that finds applications in countless downstream computer vision and graphics tasks. While many methods rely on data priors learned from large fully labeled datasets, these priors can fail to generalize to unseen shapes especially under very sparse unoriented inputs Chen et al. (2023a). Hence, it is important to design learning frameworks that can lead to efficient and robust learning of implicit shape representations under such extreme constraints.

In this context, the learning strategy introduced by Ma et al. (2021) named NeuralPull have shown to be one of the most successful ones in learning implicit shapes from point cloud unsupervisedly. However, upon observing the behavior of the training and validation errors of this method under sparse and dense input point clouds (Figure 1), we notice that the validation error starts increasing quite early on in the training in the sparse input case, while the training loss keeps on decreasing. This suggests an overfitting problem that evidently intensifies in the sparse setting. In the case where this method did not entirely diverge from the very beginning, we observe the validation chamfer gap (Figure 1) is synonymous to qualitative deterioration in the extracted shape with symptoms varying between shapes including shape hallucinations, missing shapes, and shape becoming progressively wavy, bumpy and noisy. In extreme instances, shapes break into separate components or clusters around input points. When the input is additionally noisy, this exacerbates the overfitting effect on the learned implicit representation.

Recent work in the field relies on various smoothness priors (*e.g.* Chen et al. (2023a); Gropp et al. (2020); Ben-Shabat et al. (2022)) to regularize the implicit shape functions and hence reduce overfitting especially in the sparse input point cloud regime. One side of the problem that remains underexplored however is how training pairs are sampled during learning, and understanding to which extent this sampling could affect performance. This is even the more an important question in our situation. In fact, while standard supervised learning uses typically data/label sample pairs, fitting implicit representation entails mapping spatial coordinates to labels or pseudo labels, where these spatial queries can be sampled uniformly or normally around the input point cloud. Whether it is due to it's sparsity or to the scanning device noise, a noise due to the displacement of the GT

perfect labels is introduced in the supervision of the query points, which can affect both the SDF function and gradient orientation. This is true whether this displacement keeps the points on the surface of the shape (sparsity) or can take them outside the surface (input noise). The network first produces a very smooth shape and when it tries to refine it, it tends to overfit to the noise present in this supervision signal. At this stage, further fitting on easy samples (predominant samples) means overfitting on this noise. The samples that can benefit the implicit representation can be drowned within easy samples.

Among the literature interested in such a problem, active learning advocates sampling based on informativeness and diversity Huang et al. (2010). New samples are queried from a pool of unlabeled data given a measure of these criteria. Informative samples are usually defined as samples that can reduce the uncertainty of a statistical model. However, several heuristics coexist as it is impossible to obtain a universal active learning strategy that is effective for any given task Dasgupta (2005). In our setting it is not clear what samples are the most informative for our implicit shape function and how to sample from the uncertainty regions of the implicit field. Recent work on distributionally robust optimization (DRO) Volpi et al. (2018), Rahimian & Mehrotra (2019) provide a mathematical framework to model uncertainty. In this framework, the loss is minimized over the worst-case distribution in a neighborhood of the observed training data distribution. As a special case, Adversarial training Madry et al. (2017) uses pointwise adversaries rather than adversarial joint perturbations of the entire training set.

Inspired by this literature, we propose to use adversarial samples to regularize the learning of implicit shape representation from sparse point clouds. We build on SDF projection minimization error loss in training. Typically query points are pre-sampled around the input point cloud to train such a method. We augment these queries with adversarial samples during training. To ensure the diversity of these additional samples, we generate them in the vicinity of original queries within locally adapted radii. These radii modulate the adversarial samples density with the input point cloud density, thus allowing us to adapt to the local specificities of the input during the neural fitting. Our adversarial training strategy, focuses on samples that can still benefit the network, which prevents the aforementioned overfitting while refining the implicit representation.

To test our idea, we devise experiments on real and synthetic reconstruction benchmarks, including objects, articulated shapes and large scenes. Our method outperforms the baseline as-well as the most related competition both quantitatively and qualitatively. We notice that our adversarial loss helps our model most in places where shape prediction is the hardest and most ambiguous, such as fine and detailed structures and body extremities. Experiments on a dense reconstruction setting show that our method can be useful in this setup as-well. Finally, as a result of our method illustrated in Figure 1, validation stabilizes and plateaus at convergence unlike our baseline, which makes it easier for us to decide the evaluation model epoch, given that evaluation errors are normally unavailable in unsupervised settings.

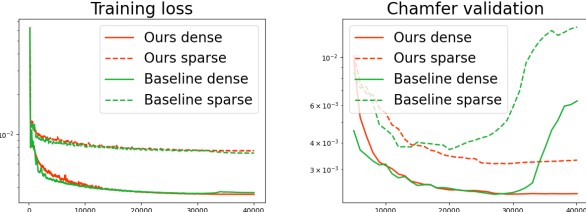

Figure 1: For shape Gargoyle of the dataset SRB, while the training loss (left) is decreasing for both our baseline (NeuralPull (NP) Ma et al. (2021)) and our method, the **Chamfer distance of reconstructions w.r.t. the GT starts increasing quite early on especially in the sparse input point cloud case for the baseline**. This undesirable behaviour is remedied by our adversarial query mining. We report Chamfer for unit box normalized meshes.

## 2 RELATED WORK

Classical shape modelling from point cloud includes combinatorical methods where the shape is defined through an input point cloud based space partitioning, through *e.g.* alpha shapes Bernardini

et al. (1999) Voronoi diagrams Amenta et al. (2001) or triangulation Cazals & Giesen (2006); Liu et al. (2020); Rakotosaona et al. (2021). Differently, the input samples can be used to define an implicit function whose zero level set represents the target shape, using global smoothing priors Williams et al. (2022); Lin et al. (2022); Williams et al. (2021) *e.g.* radial basis function Carr et al. (2001) and Gaussian kernel fitting Schölkopf et al. (2004), local smoothing priors such as moving least squares Mercier et al. (2022); Guennebaud & Gross (2007); Kolluri (2008); Liu et al. (2021), or by solving a boundary conditioned Poisson equation Kazhdan & Hoppe (2013). The recent literature proposes to parameterise these implicit functions with deep neural networks and learn their parameters with gradient descent, either in a supervised or unsupervised manner. These implicit neural representations Mescheder et al. (2019); Park et al. (2019) overcome many of the limitations of explicit ones (*e.g.* point clouds and meshes), as they allow to represent shapes with arbitrary topologies at virtually infinite resolution.

We are interested in unsupervised implicit neural shape learning. In this scenario, an MLP is typically fitted to the input point cloud without extra priors or information. Regularizations can improve the convergence and compensate for the lack of supervision. For instance, Gropp et al. (2020) introduced a spatial gradient constraint based on the Eikonal equation. Ben-Shabat et al. (2022) introduces a spatial divergence constraint. Liu et al. (2022) propose a Lipschitz regularization on the network. Ma et al. (2021) expresses the nearest point on the surface as a function of the neural signed distance and its gradient. Peng et al. (2021) proposed a differentiable Poisson solving layer that converts predicted normals into an indicator function grid efficiently. Koneputugodage et al. (2023) guides the implicit field learning with an Octree based labelling. Boulch et al. (2021) predicts occupancy fields by learning whether a dropped needle goes across the surface or no. Chen et al. (2023a) learns a surface parametrization leveraged to provide additional coarse surface supervision to the shape network. Most methods can benefit from normals if available. Atzmon & Lipman (2020) proposed to supervise the gradient of the implicit function with normals, while Williams et al. (2021) uses the inductive bias of kernel ridge regression. In the absence of normals and learning-based priors, and under input scarcity, most methods still display considerable failures. Differently from all existing methods, we explore the use of adversarial samples in training implicit neural shape representations.

In the adversarial training literature, a trade-off between accuracy and robustness has been observed empirically in different datasets Raghunathan et al. (2019), Madry et al. (2018). This has led prior work to claim that this tradeoff may be inevitable for many classification tasks Tsipras et al. (2018), Zhang et al. (2019). However, many recent papers challenged this claim. Yang et al. (2020) showed theoretically that this tradeoff is not inherent. Rather, it is a consequence of current robustness methods. These findings are corroborated empirically in many recent work Stutz et al. (2019), Xie et al. (2020), Herrmann et al. (2021).Our baseline relies on a pseudo-labeling strategy that introduces noise as the input gets sparser. Our method robustifies the learning against this noise, providing regularization and additional informative samples. The regularization helps prevent overfitting and enhances generalization ie. ensuring the loss behavior on the "training" query points is generalized in the 3D space, while informative samples aid in refining the shape function during advanced training stages.

## 3 METHOD

Given a noisy, sparse unoriented point cloud $\mathbf{P} \subset \mathbb{R}^{3 \times N_p}$, our objective is to obtain a corresponding 3D shape reconstruction, *i.e.* the shape surface $\mathcal{S}$ that best explains the observation $\mathbf{P}$. In other terms, the input point cloud elements should approximate noised samples from $\mathcal{S}$.

In order to achieve this goal, we learn a shape function $f$ parameterised with an MLP $f_\theta$. The function represents the implicit signed distance field relative to the target shape $\mathcal{S}$. That is, for a query euclidean space location $q \in \mathbb{R}^3$, $f(q) := s \cdot \min_{v \in \mathcal{S}} ||v - q||_2$, where $s := 1$ if $q$ is inside $\mathcal{S}$, and $s := -1$ otherwise. The inferred shape $\hat{\mathcal{S}}$ can be obtained as the zero level set of the SDF (signed distance function) $f_\theta$ at convergence:

$$\hat{\mathcal{S}} = \{q \in \mathbb{R}^3 \mid f_\theta(q) = 0\}. \tag{1}$$

Practically, an explicit triangle mesh for $\hat{\mathcal{S}}$ can be obtained through Marching Cubes (Lorensen & Cline (1987)) while querying neural network $f_\theta$. We note also that $\hat{\mathcal{S}}$ can be rendered through ray marching (Hart (1996)) in the SDF field inferred by $f_\theta$.

### 3.1 BACKGROUND: LEARNING A NEURAL SDF BY QUERY PROJECTION ERROR MINIMIZATION

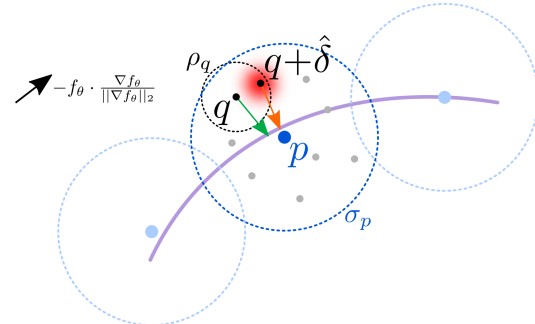

Several state-of-the-art reconstruction from point cloud methods (*e.g.* Chen et al. (2023a); Ma et al. (2022b;a); Chen et al. (2022); Ma et al. (2021)), including the state-of-the-art unsupervised reconstruction from sparse point cloud method (Chen et al. (2023a)), build on the neural SDF training procedure introduced in Ma et al. (2021) and explicitly termed Neural Pull. The latter is based on the observation that the distance field guided projection operator $q \mapsto q - f(q) \cdot \nabla f(q)$ (Chibane et al. (2020); Perry & Frisken (2001); Wolter (1993); Zhao et al. (2021)) yields the nearest surface point when applied near the surface, where $\nabla f$ is the spatial gradient of $f$.

Figure 2: We learn an implicit shape representation $f_\theta$ from a point cloud (blue points) by minimizing the error between projection of spatial queries (gray points) on the level set of the field (purple) and the nearest input point. We introduce adversarial queries to the optimization. They are defined as samples maximizing the loss in the vicinity of original queries.

In practice, query points $q \in Q$ are sampled around the input point cloud $\mathbf{P}$, specifically from normal distributions centered at input samples $\{p\}$, with locally defined standard deviations $\{\sigma_p\}$:

$$Q := \bigcup_{p \in \mathbf{P}} \{q \sim \mathcal{N}(p, \sigma_p \mathbf{I}_3)\}, \tag{2}$$

where $\sigma_p$ is chosen as the maximum euclidean distance to the $K$ nearest points to $p$ in $\mathbf{P}$. For each query $q$, the nearest point $p$ in $\mathbf{P}$ is computed subsequently, and the following objective is optimized in Ma et al. (2021) yielding a neural signed distance field $f_\theta$ whose zero level set concurs with the samples in $\mathbf{P}$:

$$\mathcal{L}(\theta, q) = ||q - f(q) \cdot \frac{\nabla f(q)}{||\nabla f(q)||_2} - p||_2^2, \quad \text{where} \quad p = \arg\min_{t \in \mathbf{P}} ||t - q||_2. \tag{3}$$

In the case of Ma et al. (2021), the training procedure is empirical risk minimization (ERM), namely:

$$\min_\theta \mathbb{E}_{q \sim Q} \mathcal{L}(\theta, q), \tag{4}$$

where $Q$ constitutes the empirical distribution over the training samples.

### 3.2 LOCAL ADVERSARIAL QUERIES FOR THE FEW SHOT SETTING

When training through standard ERM under sparse input point clouds $\mathbf{P}$, we notice the distance between the inferred shape and the ground-truth oftentimes increases and suffers from instability, even-though the training loss (over the query points $\{q\}$) can be steadily decreasing. This can be observed in Figure 1. This symptom suggests an over-fitting problem, *i.e.* the information carried out by query samples decreasing during training thus leading to poor convergence. Hence, differently from existing work in the field of learning based reconstruction from point cloud, we propose to focus on the manner in which query points $q$ are sampled at training as we hypothesise that there could be a different sampling strategy from the one proposed in Neural Pull (*i.e.* Equation 2) that can lead to a better convergence. This hypothesis is backed by literature showing that hard sample mining can lead to improved generalization and reduced over-fitting (Xie et al. (2020); Chawla (2010); Fernández-Delgado et al. (2014); Krawczyk (2016); Shrivastava et al. (2016)). Intuitively, exposing the network to the worst cases in training is likely to make it more robust and less specialized on the more common easy queries.

Hence, we explore a different procedure from standard ERM (Equation 3.1). Ideally, we wish to optimize $\theta$ under the worst distribution $Q'$ of query points $\{q\}$ in terms of our objective function, meaning:

$$\min_\theta \max_{Q'} \mathbb{E}_{q \sim Q'} \mathcal{L}(\theta, q). \tag{5}$$

Such a training procedure is akin to a distributionally robust optimization (Sagawa* et al. (2020); Rahimian & Mehrotra (2019)) which is hard to achieve in essence. It is shown however that a special more attainable case of the latter consists in harnessing hard samples locally (Staib & Jegelka (2017)), that is looking for hard samples in the vicinity of the original empirical samples:

$$\min_{\theta} \mathbb{E}_{q \sim Q} \max_{\delta, ||\delta||_2 < \rho} \mathcal{L}(\theta, q + \delta), \tag{6}$$

In order to achieve this minimization, we introduce a first order Taylor expansion of the loss:

$$\mathcal{L}(\theta, q + \delta) \approx \mathcal{L}(\theta, q) + \delta^\top \nabla_q \mathcal{L}(\theta, q). \tag{7}$$

using this approximation, we can derive the optimum value $\hat{\delta}$ as the solution to a classical dual norm problem:

$$\hat{\delta} = \rho \frac{\nabla_q \mathcal{L}(\theta, q)}{||\nabla_q \mathcal{L}(\theta, q)||_2} \tag{8}$$

where gradients $\nabla_q \mathcal{L}$ can be computed efficiently through automatic differentiation in a deep-learning framework (*e.g.* PyTorch (Paszke et al. (2019))).

We found empirically in our context that using local radii $\{\rho_p\}$ improves over using a single global radius $\rho$ and we provide and ablation of this design choice. We recall that each query point $q$ has a nearest counterpart $p$ in $\mathbf{P}$. As we want our adversarial sample $q + \hat{\delta}$ to still remain relatively close to $p$, we define $\{\rho_q\}$ as a fraction of local standard deviation $\sigma_p$ of the nearest point $p$ (*e.g.* $\rho_q = \sigma_p \times 10^{-2}$), which was used to sample query points around point $p$. We note that the definition of $\sigma_p$ is mentioned in the previous section.

To ensure the stability of our learning, we train our neural network by backpropagating a hybrid loss combining the original objective and the adversarial one, using the the strategy in Liebel & Körner (2018) for multi-task learning:

$$\mathfrak{L}(\theta, q) = \frac{1}{2\lambda_1} \mathcal{L}(\theta, q) + \frac{1}{2\lambda_2} \mathcal{L}(\theta, q + \hat{\delta}) + \ln(1 + \lambda_1) + \ln(1 + \lambda_2), \tag{9}$$

where $\lambda_1$ and $\lambda_2$ are learnable weights. A summary of our training procedure is shown in Algorithm 1. A visual illustration of our training can be seen in Figure 2.

---

**Algorithm 1** The training procedure of our method.

---

**Input:** Point cloud $\mathbf{P}$, learning rate $\alpha$, number of iterations $N_{\text{it}}$, batch size $N_b$.
**Output:** Optimal parameters $\theta^*$.
   Compute local st. devs. $\{\sigma_p\}$ $(\sigma_p = \max_{t \in K\text{nn}(p, \mathbf{P})} ||t - p||_2)$.
   $Q \leftarrow \text{sample}(\mathbf{P}, \{\sigma_p\})$ (Equ. 2)
   Compute nearest points in $\mathbf{P}$ for all samples in $Q$.
   Compute local radii $\{\rho_q\}$ $(\rho_q = \sigma_p \times 10^{-2}, \ p := \text{nn}(q, \mathbf{P}))$.
   Initialize $\lambda_1 = \lambda_2 = 1$.
   **for** $N_{\text{it}}$ times **do**
      Sample $N_b$ query points $\{q, q \sim Q\}$.
      Compute losses $\{\mathcal{L}(\theta, q)\}$ (Equ. 3)
      Compute loss gradients $\{\nabla_q \mathcal{L}(\theta, q)\}$ with autodiff.
      Compute 3D offsets $\{\hat{\delta}\}$. (Equ. 8, using radii $\{\rho_q\}$)
      Compute adversarial losses $\{\mathcal{L}(\theta, q + \hat{\delta})\}$ (Equ. 3)
      Compute combined losses $\{\mathfrak{L}(\theta, q)\}$ (Equ. 9)
      $(\theta, \lambda_1, \lambda_2) \leftarrow (\theta, \lambda_1, \lambda_2) - \alpha \nabla_{\theta, \lambda_1, \lambda_2} \Sigma_q \mathfrak{L}(\theta, q)$
   **end for**

---

## 4 RESULTS

To evaluate our method, we assess our ability to learn implicit shape representations given sparse and noisy point clouds. We use datasets from standard reconstruction benchmarks. These datasets highlight a variety of challenges of fitting coordinate based MLPs to sparse data as-well as reconstruction more broadly. Following the literature, we evaluate our method by measuring the accuracy of 3D explicit shape models extracted after convergence from our MLPs. We compare quantitatively and

qualitatively to the the state-of-the-art in our problem setting, *i.e.* unsupervised reconstruction from unoriented point cloud, including methods designed for generic pointcloud densities and methods dedicated to the sparse setting. For the former, we compare to fully implicit deep learning methods such as NP (Ma et al. (2021)), DiGs Ben-Shabat et al. (2022), NDrop (Boulch et al. (2021)), SAP (Peng et al. (2021)), IGR Gropp et al. (2020), SIREN Sitzmann et al. (2020), SAL Atzmon & Lipman (2020), PHASE Lipman (2021), in addition to hybrid methods combining implicit and grid based representations such as OG-INR Koneputugodage et al. (2023) and G-Pull Chen et al. (2023b). When it comes to methods dedicated to the sparse setting we compare to NTPS Chen et al. (2023a) which is the closest method to ours as it focuses specifically on the sparse input case. We show results for NSpline Williams et al. (2021) even-though it requires normals. We also compare to classical Poisson Reconstruction SPSR Kazhdan & Hoppe (2013). We note also that comparisons to NP (our baseline) also serves additionally as an ablation of our adversarial loss through out our experiments. For comprehensive evaluation, we also include comparisons to supervised methods including state of the art feed-forward generalizable methods, namely POCO Boulch & Marlet (2022) and CONet Peng et al. (2020) and NKSR Huang et al. (2023), alongside the fine-tuning method SAC Tang et al. (2021) and the prior-based optimization method On-Surf dedicated to sparse inputs On-Surf Ma et al. (2022a). Unless stated differently, we use the publicly available official implementations of existing methods. For sparse inputs, we experimented with point clouds of size $N_p = 1024$. While our main focus here is learning SDFs from sparse, noisy and unoriented point clouds, we found that addressing all these challenges under extremely sparse inputs (e.g. 300) leads in many cases to reconstructions that are not very meaningful or useful, and which are also hard to assess properly with standard metrics.

## 4.1 METRICS

Following seminal work, we evaluate our method and the competition w.r.t. the ground truth using standard metrics for the 3D reconstruction task. Namely, the L1 **Chamfer Distance CD$_1$** ($\times 10^2$), L2 **Chamfer Distance CD$_2$** ($\times 10^2$), the **Hausdorff distance** ($d_H$) and the euclidean distance based **F-Score (FS)** when ground truth points are available, and finally **Normal Consistency (NC)** when ground truth normals are available. We detail the expressions of these metrics in the supplementary material.

## 4.2 DATASETS AND INPUT DEFINITIONS

**ShapeNet** (Chang et al. (2015)) consists of various instances of 13 different synthetic 3D object classes. We follow the train/test splits defined in Williams et al. (2021). We generate noisy input point clouds by sampling 1024 points from the meshes and adding Gaussian noise of standard deviation 0.005 following the literature (*e.g.* Boulch & Marlet (2022); Peng et al. (2020)). For brevity we show results on classes Tables, Chairs and Lamps.

**Faust** (Bogo et al. (2014)) consists of real scans of 10 human body identities in 10 different poses. We sample sets of 1024 points from the scans as inputs.

**3D Scene** (Zhou & Koltun (2013)) contains large scale complex real world scenes obtained with a handheld commodity range sensor. We follow Chen et al. (2023a); Jiang et al. (2020); Ma et al. (2021) and sample our input point clouds with a sparse density of 100 per m$^2$, and we report performance similarly for scenes Burghers, Copyroom, Lounge, Stonewall and Totempole.

**Surface Reconstruction Benchmark (SRB)** (Williams et al. (2019)) consists of five object scans, each with different challenges such as complex topology, high level of detail, missing data and varying feature scales. We sample 1024 points from the scans for the sparse input experiment, and we also experiment using the dense inputs.

## 4.3 OBJECT LEVEL RECONSTRUCTION

We perform reconstruction of ShapeNet Chang et al. (2015) objects from sparse and noisy point clouds. Table 4.3 and Figure 4 show respectively a numerical and qualitative comparison to the competition. We outperform the competition across all metrics, as witnessed by the visual superiority of our reconstructions. We manage to recover fine structures and details with more fidelity. Although it obtains overall good coarse reconstructions, the thin plate spline smoothing prior of

NTPS seems to be hindering its expressivity. We noticed OG-INR fails to converge to satisfactory results under the sparse and noisy regime despite its effective Octree based sign field guidance in dense settings.

|  | CD1 | CD2 | NC | FS |
|---|---|---|---|---|
| SPSR | 2.34 | 0.224 | 0.74 | 0.50 |
| OG-INR | 1.36 | 0.051 | 0.55 | 0.55 |
| NP | 1.16 | 0.074 | 0.84 | 0.75 |
| Grid-Pull | 1.07 | 0.032 | 0.70 | 0.74 |
| NTPS | 1.11 | 0.067 | **0.88** | 0.74 |
| Ours | **0.76** | **0.020** | 0.87 | **0.83** |

Table 1: ShapeNet Chang et al. (2015) reconstructions from sparse noisy unoriented point clouds.

|  | CD1 | CD2 | NC | FS |
|---|---|---|---|---|
| POCO | 0.308 | 0.002 | 0.934 | 0.981 |
| Conv-onet | 1.260 | 0.048 | 0.829 | 0.599 |
| On-Surf | 0.584 | 0.012 | 0.936 | 0.915 |
| SAC | 0.261 | 0.002 | 0.935 | 0.975 |
| NKSR | 0.274 | 0.002 | 0.945 | 0.981 |
| SPSR | 0.751 | 0.028 | 0.871 | 0.839 |
| Grid-Pull | 0.495 | 0.005 | 0.887 | 0.945 |
| NTPS | 0.737 | 0.015 | 0.943 | 0.844 |
| NP | 0.269 | 0.003 | 0.951 | 0.973 |
| Ours | **0.220** | **0.001** | **0.956** | **0.981** |

Table 2: Faust (Bogo et al. (2014)) reconstructions from sparse noisy unoriented point clouds.

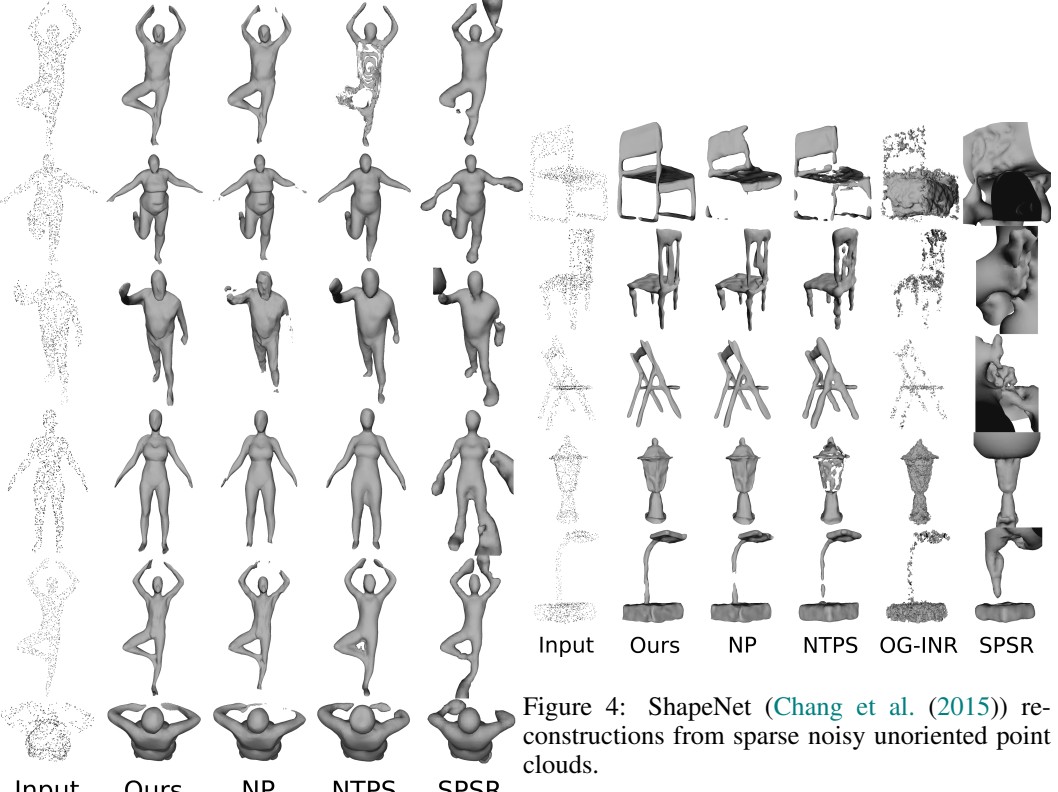

Figure 4: ShapeNet (Chang et al. (2015)) reconstructions from sparse noisy unoriented point clouds.

Figure 3: Fasut (Bogo et al. (2014)) reconstructions from sparse noisy unoriented point clouds.

## 4.4 REAL ARTICULATED SHAPE RECONSTRUCTION

We perform reconstruction of Faust (Chang et al. (2015)) human shapes from sparse and noisy point clouds. Table 4.3 and Figure 3 show respectively a numerical and qualitative comparison to the competition. We outperform the other methods across all metrics. Visually, our reconstructions are particularly better at body extremities. Similarly to fine structures in the ShapeNet experiment, these

are areas where the input point cloud samples are scarce and shape prediction is hard and ambiguous. NTPS reconstructions tend to be coarser and less detailed on this data as-well.

## 4.5 REAL SCENE LEVEL RECONSTRUCTION

Following Chen et al. (2023a), we report reconstruction results on the 3D Scene (Zhou & Koltun (2013)) data from spatially sparse point clouds. Table 3 summarizes numerical results. We compiled results for methods NTPS, NP, SAP, NDrop and NSpline as reported in state-of-the-art method NTPS. We outperform the competition is this benchmark thanks to our loss, as our baseline NP displays more blatant failures in this large scale sparse setup. Figure 5 shows qualitative comparisons to our baseline NP and SPSR. Red boxes highlights area where our method displays particular better details an fidelity in the reconstruction.

| | Burghers | | | Copyroom | | | Lounge | | | Stonewall | | | Totemple | | | Mean | | |
|---|---|---|---|---|---|---|---|---|---|---|---|---|---|---|---|---|---|---|
| | CD1 | CD2 | NC | CD1 | CD2 | NC | CD1 | CD2 | NC | CD1 | CD2 | NC | CD1 | CD2 | NC | CD1 | CD2 | NC |
| SPSR | 0.178 | 0.205 | 0.874 | 0.225 | 0.286 | 0.861 | 0.280 | 0.365 | 0.869 | 0.300 | 0.480 | 0.866 | 0.588 | 1.673 | 0.879 | 0.314 | 0.602 | 0.870 |
| NDrop | 0.200 | 0.114 | 0.825 | 0.168 | 0.063 | 0.696 | 0.156 | 0.050 | 0.663 | 0.150 | 0.081 | 0.815 | 0.203 | 0.139 | 0.844 | 0.175 | 0.089 | 0.769 |
| NP | 0.064 | 0.008 | 0.898 | 0.049 | 0.005 | 0.828 | 0.133 | 0.038 | 0.847 | 0.060 | 0.005 | 0.910 | 0.178 | 0.024 | 0.908 | 0.097 | 0.016 | 0.878 |
| SAP | 0.153 | 0.101 | 0.807 | 0.053 | 0.009 | 0.771 | 0.134 | 0.033 | 0.813 | 0.070 | 0.007 | 0.867 | 0.474 | 0.382 | 0.725 | 0.151 | 0.100 | 0.797 |
| Nspline | 0.135 | 0.123 | 0.891 | 0.056 | 0.023 | 0.855 | 0.063 | 0.039 | 0.827 | 0.124 | 0.091 | 0.897 | 0.378 | 0.768 | 0.892 | 0.151 | 0.209 | 0.88 |
| NTPS | 0.055 | 0.005 | **0.909** | 0.045 | 0.003 | **0.892** | 0.129 | 0.022 | **0.872** | 0.054 | 0.004 | **0.939** | 0.103 | 0.017 | 0.935 | 0.077 | 0.010 | **0.897** |
| Ours | **0.051** | **0.006** | 0.881 | **0.037** | **0.002** | 0.833 | **0.044** | **0.011** | 0.862 | **0.035** | **0.003** | 0.912 | **0.042** | **0.002** | 0.925 | **0.041** | **0.004** | 0.881 |

Table 3: 3D Scene (Zhou & Koltun (2013)) reconstructions from sparse point clouds.

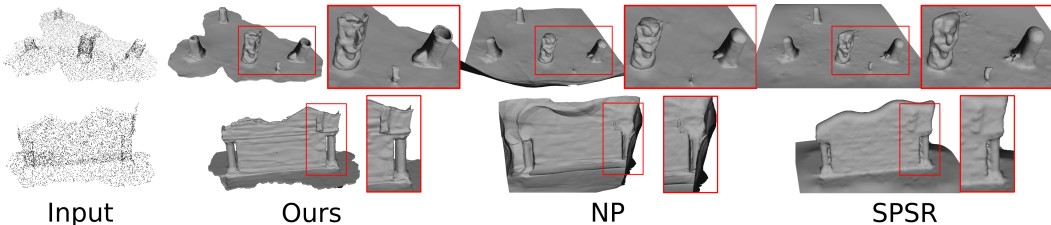

Figure 5: 3D Scene (Zhou & Koltun (2013)) reconstructions from sparse unoriented point clouds.

## 4.6 VARYING THE POINT CLOUD DENSITY

We use the SRB (Williams et al. (2019)) benchmark to assess the behavior of our method across different point cloud densities. Table 5 indicates comparative results under 1024 sized input point clouds, while Table 4 presents results under dense inputs. We compiled results for the competition from OG-INR in the dense setting. We outperform our competition in the sparse case, and we perform on par with the state-of-the-art in the dense case. Our improvement w.r.t. our baseline is substantial for both sparse and dense inputs. This can be seen visually in the supplementary material, where we show reconstructions for both sparse and dense cases. Notice how we recover better topologies in the sparse case and improved and more accurate details in the dense case, as pinpointed by the red boxes. These results showcase the utility and benefit of our contribution even in the dense setting. We note that SPSR becomes a very viable contestant qualitatively in the dense setting.

## 4.7 ABLATION STUDIES

| | CD1 | NC |
|---|---|---|
| NP (baseline) | 1.10 | 0.85 |
| Ours (local) $\rho_q = \sigma_p/10$ | 0.92 | 0.86 |
| Ours (local) $\rho_q = \sigma_p/100$ | **0.75** | 0.86 |
| Ours (local) $\rho_q = \sigma_p/500$ | 0.88 | 0.86 |
| Ours (local) $\rho_q = \sigma_p/1000$ | 1.02 | 0.84 |
| Ours (global ) $\rho = \sigma_p/100$ | 0.98 | 0.86 |

Table 6: Ablation of hard sample search radii.

|                | $d_C$ | $d_H$ |
|----------------|-------|-------|
| Nrml Est + SPSR | 1.25 | 22.59 |
| IGR wo n       | 1.38  | 16.33 |
| SIREN wo n     | 0.42  | 7.67  |
| SAL            | 0.36  | 7.47  |
| IGR + FF       | 0.96  | 11.06 |
| PHASE+FF       | 0.22  | 4.96  |
| DiGS           | **0.19** | **3.52** |
| SAP            | 0.21  | 4.51  |
| OG-INR         | 0.20  | 4.06  |
| NP             | 0.23  | 4.46  |
| OURS           | **0.19** | 3.84  |

Table 4: SRB (Williams et al. (2019)) reconstructions from dense unoriented point clouds.

|         | $d_C$ | $d_H$ |
|---------|-------|-------|
| SPSR    | 2.27  | 21.1  |
| DiGs    | 0.68  | **6.05** |
| OG-INR  | 0.85  | 7.10  |
| NTPS    | 0.73  | 7.78  |
| NP      | 0.58  | 8.90  |
| Ours    | **0.49** | 8.17 |

Table 5: SRB (Williams et al. (2019)) reconstructions from sparse unoriented point clouds.

**Perturbation radii**. The ablation of our main contribution is present through out all Tables and Figures. In fact while we use the combined loss in Equation 9, our baseline (*i.e.* NP) uses solely the query projection loss in Equation 3. The improvement brought by our additional loss is present across real/synthetic, scenes/objects, sparse/dense point clouds. Additionally, we perform an ablation of using local *vs.* global radii $\rho$ (Equation 8) and the choice of value of local radii $\rho_q$ in Table 6. Results show that using local radii is a superior strategy as it intuitively allows for a spatially adaptive search of hard samples. We note that our baseline NP constitutes the special case $\rho = 0$. A value of $\sigma_p \times 10^{-2}$ achieves empirically satisfactory results ($p$ being the nearest point to the query in the input point cloud). Decreasing $\rho_q$ leads expectedly to worse results as less hard queries are available for sampling. However, we also note that very large values of $\rho_q$ can lead to spurious pseudo supervision, as adversarial samples $q + \delta$ run the risk of no longer having the same nearest point in the input point cloud as their original sample $q$.

**Multitask loss**: To guarantee the stability of our learning process, we employ a hybrid loss that merges the original objective with an adversarial one. This approach becomes crucial when a shape-specific trade-off between adversarial regularization and the original loss is required for the convergence of the shape function. In practical terms, this strategy outperformed using the adversarial loss alone, leading to an improvement in CD1 from 0.78 to 0.75.

**Increasing the number of query points** we increase the number of NP samples to equal the total number of samples in our method (i.e. original queries + adversarial queries). We find that the performance of NP with extra queries only leads occasionally to marginal improvement. On average Chamfer distance went from 0.581 to 0.576.

## 5 LIMITATIONS

As can be seen in *e.g.* Figure 4, even-though we improve on our baseline, we still face difficulties in learning very thin and convoluted shape details, such as chair legs and slats. Although our few-shot problem is inherently challenging and can be under-constrained in many cases, we still believe there is room for improvement for our proposed strategy in this setting. For instance, one main area that could be a source of hindrance to our accuracy is the difficulty of balancing empirical risk minimization and adversarial learning. In this work, we used an off the shelf state-of-the-art self-trained loss weighting strategy, and we would like to address this challenge further as part of our future work.

## 6 CONCLUSION

We explored in this work a novel idea for regularizing implicit shape representation learning from sparse unoriented point clouds. We showed that harnessing adversarial samples locally in training can lead to numerous desirable outcomes, including superior results, reduced over fitting and easier evaluation model selection.

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
