# CAN ADVERSARIAL SAMPLES BENEFIT FEW-SHOT UNSUPERVISED IMPLICIT NEURAL SHAPE REPRESENTATION LEARNING ?
## SUPPLEMENTARY MATERIAL

## 1 DERIVATION OF EQUATION 8

The result provided in Equation 8 is derived from the following classical result: Given $x \in \mathcal{R}^3 \backslash \{0\}$,

$$\rho \frac{x}{||x||} = \arg\max_{||\delta||_2 \leq \rho} \delta^\top x$$

This can be obtained using the equality case of Cauchy–Schwarz inequality applied to the scalar product $\delta^\top x$ for $\delta$ in the a closed ball of radius $\rho$ and center 0. Then, given the perturbation $\hat{\delta}$ is defined as: $\hat{\delta} = \arg\max_{||\delta||_2 \leq \rho} \mathcal{L}(\theta, q + \delta)$, we can approximate $\hat{\delta}$ using a first order taylor expansion of the loss (Equation 6 in the paper) as:

$$\hat{\delta} \approx \arg\max_{||\delta||_2 \leq \rho} \delta^\top \nabla_q \mathcal{L}(\theta, q)$$

Consequently, the final expression follows as an application of the above result.

## 2 IMPLEMENTATION DETAILS

For sparse inputs, we experimented with point clouds of size $N_p = 1024$. While our main focus here is learning SDFs from sparse, noisy and unoriented point clouds, we found that addressing all these challenges under extremely sparse inputs (e.g. 300) leads in many cases to reconstructions that are not very meaningful or useful, and which are also hard to assess properly with standard metrics. Our MLP ($f_\theta$) follows the architecture in NP (Ma et al. (2021)). We train for $N_{it} = 40000$ iterations using the Adam optimizer. We use batches of size $N_b = 5000$. Following NP, we set $K = 51$ for estimating local standard deviations $\sigma_p$. We train on a NVIDIA RTX A6000 GPU. Our method takes 8 minutes in average to converge for a 1024 sized input point cloud. In the interest of practicality and fairness in our comparisons, we decide the evaluation epoch for all the methods for which we generated results (including our main baseline) in the same way: we chose the best epoch for all methods alike in terms of chamfer distance between the reconstruction and the input point cloud.

### METRICS

Following the definitions from Boulch & Marlet (2022) and Williams et al. (2019), we present here the formal definitions for the metrics that we use for evaluation in the main submission. We denote by $\mathcal{S}$ and $\hat{\mathcal{S}}$ the ground truth and predicted mesh respectively. All metrics are approximated with 100k samples from $\mathcal{S}$ and $\hat{\mathcal{S}}$.

,

**Chamfer Distance ($CD_1$)** The $L_1$ Chamfer distance is based on the two-ways nearest neighbor distance:

$$CD_1 = \frac{1}{2|\mathcal{S}|} \sum_{v \in \mathcal{S}} \min_{\hat{v} \in \hat{\mathcal{S}}} \|v - \hat{v}\|_2 + \frac{1}{2|\hat{\mathcal{S}}|} \sum_{\hat{v} \in \hat{\mathcal{S}}} \min_{v \in \mathcal{S}} \|\hat{v} - v\|_2.$$

**Chamfer Distance (CD$_2$)**    The L$_2$ Chamfer distance is based on the two-ways nearest neighbor squared distance:

$$\text{CD}_2 = \frac{1}{2|\mathcal{S}|} \sum_{v \in \mathcal{S}} \min_{\hat{v} \in \hat{\mathcal{S}}} \|v - \hat{v}\|_2^2 + \frac{1}{2|\hat{\mathcal{S}}|} \sum_{\hat{v} \in \hat{\mathcal{S}}} \min_{v \in \mathcal{S}} \|\hat{v} - v\|_2^2.$$

**F-Score (FS)**    For a given threshold $\tau$, the F-score between the meshes $\mathcal{S}$ and $\hat{\mathcal{S}}$ is defined as:

$$\text{FS}\left(\tau, \mathcal{S}, \hat{\mathcal{S}}\right) = \frac{2\,\text{Recall} \cdot \text{Precision}}{\text{Recall} + \text{Precision}},$$

where

$$\text{Recall}\left(\tau, \mathcal{S}, \hat{\mathcal{S}}\right) =| \left\{ v \in \mathcal{S}, \text{ s.t. } \min_{\hat{v} \in \hat{\mathcal{S}}} \|v - \hat{v}\|_2 \langle \tau \right\} |,$$
$$\text{Precision}\left(\tau, \mathcal{S}, \hat{\mathcal{S}}\right) =| \left\{ \hat{v} \in \hat{\mathcal{S}}, \text{ s.t. } \min_{v \in \mathcal{S}} \|v - \hat{v}\|_2 \langle \tau \right\} |.$$

Following Mescheder et al. (2019) and Peng et al. (2020), we set $\tau$ to 0.01.

**Normal consistency (NC)**    We denote here by $n_v$ the normal at a point $v$ in $\mathcal{S}$. The normal consistency between two meshes $\mathcal{S}$ and $\hat{\mathcal{S}}$ is defined as:

$$\text{NC} = \frac{1}{2|\mathcal{S}|} \sum_{v \in \mathcal{S}} n_v \cdot n_{\text{closest}(v,\hat{\mathcal{S}})} + \frac{1}{2|\hat{\mathcal{S}}|} \sum_{\hat{v} \in \hat{\mathcal{S}}} n_{\hat{v}} \cdot n_{\text{closest}(\hat{v},\mathcal{S})},$$

where

$$\text{closest}(v, \hat{\mathcal{S}}) = \text{argmin}_{\hat{v} \in \hat{\mathcal{S}}} \|v - \hat{v}\|_2.$$

**Hausdorff distance ($d_H$)**    This metric is defined as follows:

$$d_H = \max\left( \max_{v \in \mathcal{S}} \min_{\hat{v} \in \hat{\mathcal{S}}} \|v - \hat{v}\|_2, \max_{\hat{v} \in \hat{\mathcal{S}}} \min_{v \in \mathcal{S}} \|v - \hat{v}\|_2 \right)$$

## 3    SRB BENCHMARK VISUAL COMPARAISION IN

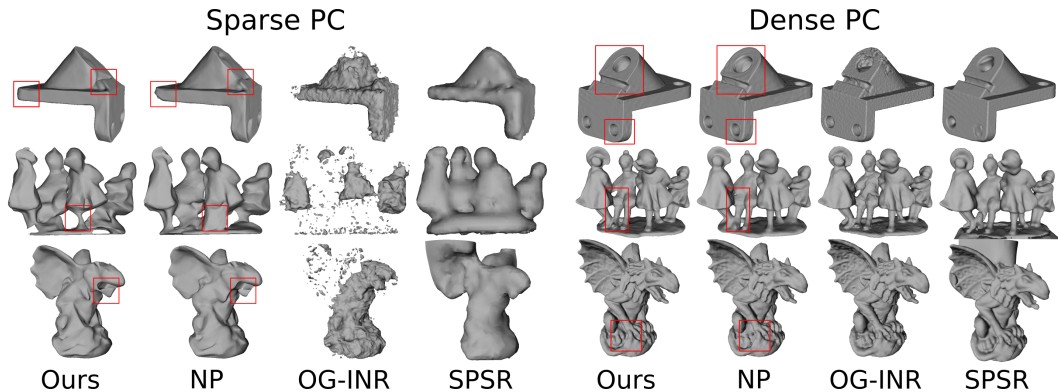

Figure 1: SRB (Williams et al. (2019)) reconstructions from unoriented sparse and dense inputs.

## 4    ADDITIONAL ABLATION

**Increasing the number of input points** we compare our method with 1024 input points to NP baseline with varying number of input points by adding 1/3 of the initial input size ( 1024) at each run.

|                        | CD1  |
|------------------------|------|
| NP + (N =1024)         | 1.10 |
| NP + 33% (N = 1365)    | 0.92 |
| NP + 66% (N = 1707)    | 0.54 |
| NP + 100% (N = 2048)   | **0.46** |
| Ours (N = 1024)        | 0.49 |

Table 1: NP Baseline performance with varying number of input points on the SRB benchmark.