# OpenReview forum: "Can adversarial samples benefit few-shot unsupervised implicit neural shape representation learning ?"
_ICLR.cc/2024/Conference — ICLR 2024 Conference Desk Rejected Submission_

### Official Review · Reviewer_ZZdy · 2023-10-24

**Soundness:** 2 fair
**Presentation:** 3 good
**Contribution:** 2 fair
**Rating:** 6
**Confidence:** 3

**Summary:**

This work addresses the problem of learning to recover SDFs from point clouds, which may be sparse or dense, without ground truth supervision. A method for mining adversarial samples around a shape’s surface is proposed, which aids in learning better neural SDFs and leads to improved robustness towards overfitting.

**Strengths:**

1) The paper is well organized and easy to follow.

2) An extensive set of experiments are conducted in which previous methods are outperformed in the sparse input case and SOTA is matched in the dense input case.

3) Qualitatively the proposed method seems to perform better than previous methods on reconstructing thin structures and fine details in the presence of low point density inputs.

**Weaknesses:**

1) Considering that the main contribution of the paper revolves around Eq. 6-8, there should be a formal derivation for $\hat{\delta}$ when using the first order Taylor expansion of loss $\mathcal{L}$. I would expect to see this in the supplemental for completeness, but couldn’t find any such derivation.

2) For sparse inputs on ShapeNet and Faust, it seems like NTPS (Chen 2023) evaluate on 300 points while here sparse inputs are being considered as 1024 points. Since NTPS is the other main work that does unsupervised SDF learning from sparse point clouds, it would be nice to see results on 300 points which are significantly sparser inputs then what has been presented.

3) Best values in Table 3 should be bolded to make it easier to read.

4) Figure 5 should include a comparison to NTPS as it performs the closest to the proposed method according to Table 3.

5) In Tables 4 and 5, metric $d_{C}$ is never defined. I assume this is chamfer distance.

6) DiGS seems to perform slightly better in the dense point cloud case (Table 4), but it wasn’t compared against in the sparse input case (Table 5). I would expect to see a comparison to this method for the sparse input to see if it performs worse. Additionally, NTPS should be included in both Table 4 and 5.

7) Please add the point cloud inputs to Figure 6 to have an understanding of the difference in inputs.

**Questions:**

1) Considering that extra query samples are generated with the proposed method, did you train the Neural Pull baseline with the same total number of samples as the proposed method or with just the same initial number of samples? It would be good to know how much more effective the approach is to just increasing the number of samples initially generated.

2) I'm a little confused on what "global radii" is in the ablation. In Table 6, it has the same definition as one of the local radii (i.e. $\rho_{q} = \sigma_{p}/100$).

3) Instead of using a hybrid loss with learnable weights, did you try just training with the adversarial loss or with a fixed weighting between the original objective and adversarial one?

4) Why is the term “few shot” used in this work? Few shot seems like it would imply this work is learning a network that generalizes to a large set of SDFs from only training with a few shapes; however, in this work a single SDF is being fit separately for each sparse point cloud. This is really unsupervised SDF learning from sparse point clouds.

---

> ### Author Response · Authors · 2023-11-22
>
> We thank the reviewer for their helpful feedback. We address below their main concerns.
>
> ### Considering that the main contribution of the paper revolves around Eq. 6-8, there should be a formal derivation for when using the first order Taylor expansion of loss . I would expect to see this in the supplemental for completeness, but couldn’t find any such derivation
> We thank the reviewer for pointing this out. We will provide the full derivation in the updated supplementary material. The result is obtained as follows:
> - The perturbation $\hat{\delta}$ is defined as $\hat{\delta}=argmax_{|| \delta||_2 \leq \rho} \mathcal{L}(\theta,q+\delta)$.
>
> - The expression provided in Equation 8 is derived from the following classical result: Given $x \in  \mathcal{R}^3 \backslash\lbrace{0}\rbrace $, $\rho  \frac{x}{||x||}=argmax_{|| \delta||_2 \leq \rho}\delta^\top x$.
>
> This can be obtained using  the equality case of Cauchy–Schwarz inequality applied to the scalar product $\delta^\top x$ for $\delta$ in the a closed ball of radius $\rho$ and center 0.
>
>
> - Using a first order taylor expansion of the loss (Equation 6 in the paper), $\hat{\delta}$ can be approximated as  $\hat{\delta} \approx argmax_{|| \delta||_2 \leq\rho}\delta^\top \nabla_q \mathcal{L}( \theta,q)$. The final expression follows as an application of the above result.
>
>
>
> ### For sparse inputs on ShapeNet and Faust, it seems like NTPS (Chen 2023) evaluate on 300 points while here sparse inputs are being considered as 1024 points. Since NTPS is the other main work that does unsupervised SDF learning from sparse point clouds, it would be nice to see results on 300 points which are significantly sparser inputs then what has been presented.
>
> While our main focus here is learning SDFs from **sparse, noisy and unoriented point clouds**, we found that addressing all these challenges under extremely sparse inputs (e.g. 300) leads in many cases to reconstructions that are not very meaningful or useful, and which are also hard to assess properly with standard metrics. This can be seen in some of the results presented in the paper NTPS as well as in the ShapeNet examples presented [here](https://anonymous.4open.science/r/Adv-F74C/ShapeNet/lamp_300_noisy/ours_ntps.mp4). We will add visual examples in the next version of our supplementary material to illustrate this point.
>
> In particular, we note also that differently from NTPS, and in order to align with standard ShapeNet reconstruction benchmarks (e.g. CONet Peng et al. 2020), our ShapeNet experiment features noisy point clouds (with standard deviation 0.005) and not clean ones as in NTPS. In this case, reconstructing from noisy 300 sized inputs often leads to catastrophic meshes, and it is hard to conceive that such reconstructions can be of purpose or evaluated fairly and properly. Hence, we judged that under heavy input noise, a more reasonable objective for our unsupervised learning would be with slightly denser inputs.
>
> **We note that in the 3D scene reconstruction benchmark, we found the spatial density of the input to be reasonable and the noise not very high. Hence, we experimented with the exact same input density as NTPS and we reported results for the competition as shown in the latter.**
>
> ### Best values in Table 3 should be bolded to make it easier to read.
>
>  Thank you for the advice. We will update Table 3 accordingly.

---

> ### Author Response · Authors · 2023-11-22
>
> ### Figure 5 should include a comparison to NTPS as it performs the closest to the proposed method according to Table 3.
>
> Using their publicly available implementation, we managed to obtain results for object reconstruction benchmarks ShapeNet, Faust and SRB for NTPS. However, we could not reproduce the paper results in the 3D Scene setting using this same implementation. We think this benchmark might require some special hyperparameter setting (e.g. a specific number of patches, optimizer parameters, etc) that was not shared by the authors. We reached out to the authors for help and we have not received any response yet. Hence, we can only report their numerical results in this benchmark at this point.
>
> ### In Tables 4 and 5, metric is never defined. I assume this is chamfer distance.
>
> We apologize for this mix-up. It is indeed chamfer. We will correct this in the next version.
>
> ### DiGS seems to perform slightly better in the dense point cloud case (Table 4), but it wasn’t compared against in the sparse input case (Table 5). I would expect to see a comparison to this method for the sparse input to see if it performs worse. Additionally, NTPS should be included in both Table 4 and 5.
>
> We thank the reviewer for raising this point. We agree indeed that comparison to NTPS and DIGS in the sparse SRB benchmark could position our method better with respect to existing work. Consequently, we run these methods under the sparse SRB experiment. We will report the updated Table 4 in the next version of the paper, and we show these new results here below as well. We outperform both NTPS and DIGS in the sparse SRB setting thanks to our adversarial sample mining strategy. We note that NTPS failed to converge to reasonable reconstructions in the dense SRB setting. We believe that it is due to the fact that its hyperparameters as provided in the public implementation are designed and tuned specifically for sparse point cloud density. We note that  no results are reported in the paper on dense point clouds neither.
>
> |        | Sparse   |          | Dense |  |
> |--------|---------------|-------------|-------|-------|
> |        | CD            | HD          | CD    | HD    |
> | SPSR   | 2.27          | 21.1        | 1.25  | 22.59 |
> | DiGs   | 0.68          | **6.05** |**0.19**  | **3.52**  |
> | OG-INR | 0.85          | 7.10        | 0.20  | 4.06  |
> | NTPS   | 0.73          | 7.78        | -     | -     |
> | N-Pull | 0.58          | 8.90        | 0.23  | 4.46  |
> | Ours   | **0.49** | 8.17        | **0.19**  | 3.84  |
> Table 5: SRB  reconstructions from sparse noisy unoriented point clouds.
>
>
> ### Please add the point cloud inputs to Figure 6 to have an understanding of the difference in inputs.
> We thank the reviewer for the remark. We will update Figure 6 with the input point clouds as requested. We note that the videos that we provide at the request of reviewer PU6C show these sparse point clouds.

---

> ### Author Response · Authors · 2023-11-22
>
> **Questions:**
> ### Considering that extra query samples are generated with the proposed method, did you train the Neural Pull baseline with the same total number of samples as the proposed method or with just the same initial number of samples? It would be good to know how much more effective the approach is to just increasing the number of samples initially generated
>
>
> This is an interesting point. Following the request of the reviewer we perform the experiment where we increase the number of NP samples to equal the total number of samples in our method (I.e. original queries + adversarial queries). We find that the performance of NP with extra queries only leads occasionally to marginal improvement. **On average Chamfer distance went from 0.581 to 0.576**. Our adversarial sample mining strategy outperforms both the baseline and increasing the size of queries through naïve normal upsampling.  This is an expected result as **harnessing adversarial queries allows us to mine the most informative samples**, which are usually drowned within the ordinary easy samples. Increasing the size of samples does not guaranty access to a high rate of informative ones. We will this result and  a discussion in the next version of paper.
>
> ### I'm a little confused on what "global radii" is in the ablation. In Table 6, it has the same definition as one of the local radii
>
> We apologize for the mix-up. Global radius means using one single search radius for all query points alike. Local radii means defining a different radius per query point, which is linked to the local density of the input point cloud as explained in section 3.2. We will clarify this.
>
> ###  Instead Instead of using a hybrid loss with learnable weights, did you try just training with the adversarial loss or with a fixed weighting between the original objective and adversarial one?
>
> Following the request of the reviewer, we provide an additional ablation comparing the hybrid loss with learnable weights to the adversarial loss alone (Ours w/o learnable weights).
> |                            | CD1  | NC   |
> |----------------------------|------|------|
> | Ours w/o learnable weights | 0.78 | 0.86 |
> | Ours                       | 0.75 | 0.86 |
> Table: Ablation of loss function.
> ### Why is the term “few shot” used in this work? Few shot seems like it would imply this work is learning a network that generalizes to a large set of SDFs from only training with a few shapes; however, in this work a single SDF is being fit separately for each sparse point cloud. This is really unsupervised SDF learning from sparse point cloud
>
>
> We apologize for the confusion. We consider the input point cloud samples as training samples. We only observe a few of them in this setting, as opposed to the task of reconstruction from dense point cloud, hence the denomination “few shot”. A similar terminology is used in recent literature learning NeRF from sparse images of a scene without data priors (e.g. CVPR23 Improving Few-shot Neural Rendering with Free Frequency Regularization). We agree that unsupervised learning from sparse point cloud is also a well fitted description for our work and we have no issue renaming out paper accordingly if the reviewers agree about this.

---

> ### Comment · Reviewer_ZZdy · 2023-11-22
>
> Thank you for the detailed response. Please make the changes mentioned here in the next version of the paper, especially adding the derivation of your proposed loss as well as the motivation for not using sparser inputs like were used in NTPS.
>
> Most of my concerns have been addressed and questions have been answered; however, I do have one more question. In the sparse setting of the SRB benchmark (Table 5), do you have some insight as to why the proposed approach obtains better chamfer distance yet worse Hausdorff distance than several methods (i.e., DiGs, OG-INR, NTPS)? Particularly, OG-INR qualitatively has significantly worse reconstructions and is highly fragmented (in Figure 6), so there should be points in the ground truth shape whose distance to its nearest neighbor in the reconstruction is very large. On the other hand, the proposed method seems to produce overly smoothed results for the sparse setting. To me, it seems like the former of the two would observe larger Hausdorff distance though, but the results in Table 5 suggest otherwise.

---

> > ### Author Response · Authors · 2023-11-22
> >
> > Thank you very much for acknowledging our rebuttal.
> > We note that smoothness in the sparse SRB reconstructions is in line with the very low input density compared to the high level of detail in the ground-truth shapes. The input point clouds can be seen in the video uploaded at the request of reviewer PU6C.
> > HD is very sensitive to outliers. We think the second order direct implicit function regularization in DIGS and NTPS can lead to high robustness against these. OG-INR grid based supervision can also be effective against outliers as any false positives far away from the surface are dismissed de facto. These are all direct regularizations on the SDF. Differently, ours is a regularization acting on the loss spatial landscape. While it leads to more accurate reconstructions (CD), it could be not as effective against outliers in some cases. We note that outliers can also be filtered relatively easily as a post process.

---

> > > ### Comment · Reviewer_ZZdy · 2023-11-23
> > >
> > > Thank you for the explanation in regards to robustness to outliers of previous methods, albeit it still isn't quite clear to me how the obviously worse reconstructions from OG-INR in Figure 6 is getting better HD in Table 5. If HD is being computed from points sampled on the ground truth and reconstructed meshes, the missed geometry in the reconstruction should produce very high HD.
> > >
> > > With that being said, the authors have still managed to more or less address all my major concerns/weaknesses and therefore I have updated my score to reflect this.

---

### Official Review · Reviewer_iVm1 · 2023-10-29

**Soundness:** 3 good
**Presentation:** 2 fair
**Contribution:** 3 good
**Rating:** 5
**Confidence:** 4

**Summary:**

Neural Signed Distance Functions (SDF) is a power implicit representation of 3D shapes. However, to robustly learn such a representation from sparse point cloud is a challenging problem. The paper introduces leverages adversarial samples around the shape to improve the learned SDFs. Specifically, it first compute the loss per sampled query point (Eq.4) and then augment the sampled query point set by selecting additional query points which maximize the computed loss (Eq.6). The radii of the range in which the adversarial query points are sampled is adaptively changed based on statistics per local region.

The 3D reconstruction experiments are conducted on four datasets covering 3D objects, 3D human bodies, and 3D scenes. For each dataset, the proposed method is compared against at least three recent approaches, and achieves the best performance in most cases under different evaluation metrics (except for Normal Consistency (NC) for 3D scene). Qualitative results of each dataset are also shown for comparison. The ablation study compares the effect of different values of adversarial sample are radii.

**Strengths:**

1. $\textbf{Problem formulation}$: neural signed distance function (sdf) is indeed an important representation of 3D shapes, and how to learn such a representation robustly without over-fitting is an interesting and useful topic, especially under sparse point cloud cases. This paper tried to tackle an important research problem.

2. $\textbf{Method soundness}$. The proposed adversarial sampling strategy is sound and the math formulas are valid.

3. $\textbf{Experimental results}$. The performance is superior than the compared approaches in most cases, especially for $L_1$ Chamfer Distance (CD1) and $L_2$ Chamfer Distance (CD2).

**Weaknesses:**

1. $\textbf{Method}$

1.1 The proposed strategy to find adversarial query points is an extension of Fast Gradient Sign Method Adversarial Training (FGSM-AT) [1] and Projected Gradient Descent (PGD) [2] from 2D image classification to 3D point cloud sample, with the former trying to find adversarial images and this paper trying to find adversarial points.

1.2 Using Taylor expansion to estimate the loss value at a certain point has also been proposed in previous literatures [3], [4].

Though applying adversarial training to SDF learning is interesting and somewhat novel, an in-depth analysis of relating SDF fields to 3D shape geometry is welcome to strength the paper.

2. $\textbf{Experiments}$

2.1 It is confusing in Table 3 that CD1 and CD2 outperforms all the compared methods, but NC is inferior to other methods. If a justification could be provided or the performance in NC metric could be improved, the paper will be more strengthened.

2.2 The proposed method is used to sample additional adversarial points to augment the queried points, what if I uniformly (or randomly) sample more points (or use Gaussian distribution to sample more points), instead of adversarially sampling as augmentation? If such an experiments can be presented, it will lend strong support to the effectiveness of the adversarial sampling.

2.3 Another suggestive ablation experiment to strengthen the paper is to compare the proposed approach on point clouds with 1024 points with a baseline approach (i.e., without adversarial sampling) on point clouds with more than 1024 points (like 2048 points).

3. $\textbf{Writting}$

3.1 This paper needs a more careful proof-reading and grammar checking. For example, "using this approximation" (above Eq.8) should be "Using this approximation".

3.2 The legend text in Figure 1 is too small to be seen clearly.

3.3 It is preferable to add citations in Tables to make it easier to check the reference papers.

3.4 Figure 3 and Figure 4 should be combined as a single big figure to avoid large empty space.

$\newline$

[1]. Ian J. Goodfellow, Jonathon Shlens, and Christian Szegedy. Explaining and harnessing adversarial examples. ICLR, 2015.

[2]. Aleksander Madry, Aleksandar Makelov, Ludwig Schmidt, Dimitris Tsipras, and Adrian Vladu. Towards deep learning models resistant to adversarial attacks. ICLR, 2018.

[3]. Jin G, Yi X, Wu D, Mu R, Huang X. Randomized adversarial training via taylor expansion. CVPR, 2023.

[4]. Qian YG, Zhang XM, Swaileh W, Wei L, Wang B, Chen JH, Zhou WJ, Lei JS. TEAM: An Taylor Expansion-Based Method for Generating Adversarial Examples. arXiv preprint arXiv:2001.08389. 2020.

**Questions:**

Please refer to the weakness part. If the weaknesses can be resolved, I will raise my rating.

---

> ### Author Response · Authors · 2023-11-20
>
> We thank the reviewer for their constructive feedback. We address below their main comments.
>
> ### 1.1 The proposed strategy to find adversarial query points is an extension of Fast Gradient Sign Method Adversarial Training (FGSM-AT) [1] and Projected Gradient Descent (PGD) [2] from 2D image classification to 3D point cloud sample, with the former trying to find adversarial images and this paper trying to find adversarial points.
> ### 1.2 Using Taylor expansion to estimate the loss value at a certain point has also been proposed in previous literatures [3], [4].
> ### Though applying adversarial training to SDF learning is interesting and somewhat novel, an in-depth analysis of relating SDF fields to 3D shape geometry is welcome to strength the paper.
>
> We kindly remind that, as the reviewer referred to it, our PGD-like optimization concerns the query points of our coordinate-based MLP and not the input point cloud. Consequently, the distinction between natural and adversarial images in 2D classification  can’t be extended to our setting as the network is expected to perform well uniformly in the 3D space including eventual“spatial adversaries”.
>
> Our baseline utilizes  pseudo-labels generated by pulling queries to their nearest point cloud samples. The denser the point cloud, the cleaner the pseudo-labels. We  address the sparsity of the point cloud, likening it to the noise introduced by the pseudo-label supervision when the input gets sparser.
>
> As can be seen in the uploaded videos, when the network fails in some regions of the 3D space, it results in artifacts, bumps (see Gargoyle   [uploaded video](https://anonymous.4open.science/r/Adv-F74C/srb/gargoyle/NP.mp4)  ) and missing parts (like body extremities in Faust in Figure 4, and the floor in the Scene copy room of the [uploaded video](https://anonymous.4open.science/r/Adv-F74C/3dscene/copyroom/NP.mp4) ) in the 3D shape geometry.  This means some points,  dubbed  « spatial adversaries », are pulled far away from their associated input points. While the loss in these points can be high, the average loss is dominated by “easy samples” at advanced stages of the training.
>
> Our method robustifies the learning against this noise, providing regularization and additional informative samples. The **regularization** helps prevent overfitting and enhances generalization ie. ensuring the loss behavior on the “training” query points is generalized to “spatial adversaries”, while **informative samples** aid in refining the shape function during advanced training stages.

---

> ### Author Response · Authors · 2023-11-20
>
> ### 2.1 It is confusing in Table 3 that CD1 and CD2 outperforms all the compared methods, but NC is inferior to other methods. If a justification could be provided or the performance in NC metric could be improved, the paper will be more strengthened.
>
> Firstly, we note that in Table 3, we compile results for the competition from the paper NTPS (Chen et al. 2023). Our training with spatial adversaries acts as a regularization to our learning, through the spatial smoothing of the reconstruction loss. We believe that **excess regularization can lead to overly smooth surfaces in some cases**, and that this explains in turn our normal consistency numbers. This can be improved through tuning the hyperparameters of our method, by **finding the best tradeoff between our two losses** for each benchmark or scene separately, and we did not explore intensively yet. We will provide a discussion of this element in our limitation section of the next version of the manuscript.
>
> ### 2.2 The proposed method is used to sample additional adversarial points to augment the queried points, what if I uniformly (or randomly) sample more points (or use Gaussian distribution to sample more points), instead of adversarially sampling as augmentation? If such an experiments can be presented, it will lend strong support to the effectiveness of the adversarial sampling.
>
> This is an interesting point. Following the request of the reviewer we perform the experiment we increase the number of NP samples to equal the total number of samples in our method (I.e. original queries + adversarial queries). We find that the performance of NP with extra queries only leads occasionally to marginal improvement. **On average Chamfer distance went from 0.581 to 0.576**.  Our adversarial sample mining strategy outperforms both the baseline and increasing the size of queries through naïve normal upsampling.  This is an expected result as **harnessing adversarial queries allows us to mine the most informative samples**, which are usually drowned within the ordinary easy samples. Increasing the size of samples does not guaranty access to a high rate of informative ones. We will this result and  a discussion in the next version of paper.
>
> ### 2.3 Another suggestive ablation experiment to strengthen the paper is to compare the proposed approach on point clouds with 1024 points with a baseline approach (i.e., without adversarial sampling) on point clouds with more than 1024 points (like 2048 points).
>
> Following the request of the reviewer, we compare our method with 1024 input points to NP baseline with varying number of input points by adding 1/3 of the initial input size ( 1024)  at each run.
>
> |                      | CD1     |
> |----------------------|---------|
> | NP + (N =1024)      | 0.58  |
> | NP + 33% (N = 1365)  | 0.57    |
> | NP + 66% (N = 1707)  | 0.54  |
> | NP + 100% (N = 2048)  | 0.46 |
> | Ours (N = 1024)      |     0.49    |
> Table: NP Baseline performance with varying number of input points on the SRB benchmark.

---

### Official Review · Reviewer_akZJ · 2023-10-29

**Soundness:** 2 fair
**Presentation:** 2 fair
**Contribution:** 3 good
**Rating:** 6
**Confidence:** 5

**Summary:**

This paper attempts to enhance the generalization of reconstructing meshes from sparse point clouds by employing an optimization strategy similar to adversarial training. The authors observed that as the input point cloud becomes sparser, the phenomenon of network overfitting becomes more severe during the experiments with the NeuralPull method. Therefore, they propose the need to mitigate this overfitting phenomenon using some strategy. The authors suggest not directly optimizing randomly sampled points in space, but instead identifying the worst-performing points in space (defined as points with the highest loss value) and then aligning the implicit function with the given point cloud as closely as possible at these identified points. Technically, the authors employ a method similar to the PGD attack to train these worst-performing points. Finally, the authors conduct experiments on both object and scene point cloud data, and in comparison, achieve favorable results in terms of experimental outcomes.

**Strengths:**

The article attempts to address an important problem and introduces relevant concepts from other fields to solve it (such as defining overfitting as a flaw in the sampling strategy, leading to the introduction of a strategy similar to adversarial training). Additionally, the paper compares different methods and experimentally demonstrates the effectiveness of the proposed approach. Overall, this is not an incremental work, as it introduces novel ideas and approaches to tackle the problem.

**Weaknesses:**

However, I do have some concerns regarding the proposed method in this paper.

Firstly, from my perspective, using a PGD-like optimization method imposes a smooth constraint on the implicit function (considering that the Lipschitz constant of the model generally decreases after adversarial training). While this approach is indeed an effective regularization for noisy point clouds, it may not be as effective for sparse point clouds. We cannot deny the existence of sparse surfaces with high curvature, and further experiments may be needed to address this issue.

Secondly, I am unsure how the authors obtained the normals of the point cloud. Screened Poisson Surface Reconstruction (SPSR) requires this parameter, and it heavily relies on the consistency of normal directions. If the normals are estimated incorrectly, it could introduce unfairness in the comparison with SPSR. More experimental details need to be provided, and at least the optimization of results using normal flipping strategies should be considered.

Furthermore, there is a significant amount of related work in this field that has not been compared. I suggest the authors refer to [1] and provide further and more comprehensive comparisons.

Lastly, it is generally believed that the performance on the original samples tends to decrease after adversarial training. It would be helpful to provide additional explanations as to why this performance decrease is reasonable.

[1] Huang Z, Wen Y, Wang Z, et al. Surface reconstruction from point clouds: A survey and a benchmark. arXiv preprint arXiv:2205.02413, 2022.

**Questions:**

As mentioned above, I see the potential in this paper, but the current version still has some way to go before it can be accepted by a top conference. At the very least, the following issues need to be addressed:

1. Carefully examine the scope of the method to ensure there is no overclaiming.

2. Provide additional experimental details to enable fairer comparisons between the proposed method and others, and include more relevant experiments.

3. If possible, provide further theoretical derivations to explain what adversarial training specifically improves in the implicit function.

If my concerns mentioned above are addressed, I would consider raising the score.

---

> ### Author Response · Authors · 2023-11-20
>
> We thank the reviewer and appreciate their feedback. We address below their main concerns.
>
> ### Firstly, from my perspective, using a PGD-like optimization method imposes a smooth constraint on the implicit function (considering that the Lipschitz constant of the model generally decreases after adversarial training). While this approach is indeed an effective regularization for noisy point clouds, it may not be as effective for sparse point clouds. We cannot deny the existence of sparse surfaces with high curvature, and further experiments may be needed to address this issue.
>
> **Why is our strategy effective with sparsity and not just noise?**
>
> We kindly remind that, as the reviewer referred to it, our PGD-like optimization concerns the query points of our coordinate-based MLP and not the input point cloud. The point cloud samples allow us to compute the “pseudo-labels” for our optimization. In fact, our network learns an SDF by pulling queries to their nearest point cloud sample. The more accurate the query-point cloud sample association, the better this strategy. Ergo, **the denser the point cloud, the “cleaner” and more precise our labels are. Consequently, the sparsity of the point cloud is analogous to noise (i.e. imprecision) in our pulling based training sample labels**.  This actually explains why our idea works, in concordance with the comment of the reviewer. We thank the reviewer for raising this point and we will clarify it further in the next version.
>
> **Do we impose a smoothness constraint on the implicit function ?**
>
> We do not believe that we impose such a constraint on the function directly. **Our method only enforces a spatial smoothing constraint on the training loss**. Hence, our constraint is indirect and **adaptive**, as it self-adjusts based on the behavior of the loss, i.e. the current local properties of the implicit function. Conversly, competition methods often impose first order (IGR (Gropp et al. 2020)) and second order (NTPS (Chen et al. 2023), DIGS (Ben-Shabat et al. 2022)) derivative constraints on the model directly, which thus could be more likely to prove ineffective in the sparse point cloud setting. We believe this to be another major difference with the existing literature that contributes to the originality of our work.
>
> ### Secondly, I am unsure how the authors obtained the normals of the point cloud. Screened Poisson Surface Reconstruction (SPSR) requires this parameter, and it heavily relies on the consistency of normal directions. If the normals are estimated incorrectly, it could introduce unfairness in the comparison with SPSR. More experimental details need to be provided, and at least the optimization of results using normal flipping strategies should be considered.
>
> Following our peer reviewed competition, we use the public established implementations for SPSR, namely [Open3D](http://www.open3d.org) and [Pymeshlab](https://pymeshlab.readthedocs.io/en/latest/index.html). We pick the best result each time out of the two. We strived to obtain the best possible outcome for these by grid searching hyper parameters, including the octree depth, the points interpolation weight, the number of points per leaf in addition to the number of nearest neighbors used in constructing the Riemannian graph used to propagate normal orientation. We note that these libraries include a normal estimation  algorithm based on local point cloud co-variance estimation and normal orientation propagation with minimum spanning trees.
>
> We note that while these implementations can perform well under relatively dense point clouds, their performance deteriorates considerably under noise, and sparsity. These two elements affect both the normal estimation stage and the boundary conditions necessary for Poisson. Deep learning based methods such as ours on the other hand can remedy such limitations by bypassing the need for point orientation as input, through modeling the reconstruction via gradient decent on highly non-linear functions.
>
> Additionally, we remind that **our numerical comparison to SPSR in Table 3 was reported in NTPS (Chen et al. 2023), and our comparison to SPSR in Table 4 was reported in OG-INR (Koneputugodage et al. 2023)**. We note also that both in Tables 3 & 4, we provide a comparison to SAP (Peng et al. 2021), a method that incorporated the Poisson regularization within a deep learning framework.

---

> > ### Comment · Reviewer_akZJ · 2023-11-22
> >
> > I thank the authors for the prompt response, but my concerns have not been completely resolved.
> >
> > 1. Indeed, I agree with the author's statement that sparsity can be considered as a form of "noise." However, my question remains unanswered. What I want to emphasize is that sparse noise and point displacement noise cannot be regarded as the same thing. Let me provide an imperfect example: sparse noise refers to the existence of numerous unobserved regions, necessitating inpainting for areas without observed points. On the other hand, point displacement noise refers to differences between the observed values of each point and their ground truth values, requiring the calculation of a delta to regress each point back to its original position. Perhaps my understanding of using adversarial training as a means of smoothing the implicit function is not accurate. Therefore, my question is, how can adversarial training simultaneously handle these two distinct types of noise?
> >
> > 2. If the author has already used MST to ensure the consistency of normals' directions, then my concern is alleviated.
> >
> > 3. Indeed, I overlooked the aspect that there are indeed numerous related works on point cloud reconstruction, making it impossible to cover them all. What I meant was that perhaps the author could classify these methods and select representative ones from each category to achieve a more comprehensive comparison with the proposed method.
> >
> > 4. Alright, I understand. I look forward to seeing these relevant discussions in the next version of the paper.

---

> ### Author Response · Authors · 2023-11-20
>
> ### Furthermore, there is a significant amount of related work in this field that has not been compared. I suggest the authors refer to [1] and provide further and more comprehensive comparisons.
>
>
> We believe that we have provided comparison to a considerable amount of sota methods using standard metrics, and covering various shape scenarios, including object (ShapeNet and SRB benchmarks), non-rigid shape (Faust Benchmark) and scene level (3D scenes benchmark) reconstruction from point cloud. We remind that we have compared to methods such as NP (Ma et al. 2021) which is our baseline, the sota method in reconstruction from extremely sparse point cloud NTPS (Chen et al. 2023) which we consider to be our mainly thematic competitor, in addition to a variety of the most recent sota general reconstruction from point cloud methods, including DIGS (Ben-Shabat et al. 2022), OG-INR (Koneputugodage et al. 2023), SAP (Peng et al. 2021), IGR (Gropp et al. 2020), SIREN (Sitzmann et al. 2020), SAL (Atzmon & Lipman 2020), PHASE (Lipman 2021), NSpline (Williams et al. 2021), NDrop (Boulch et al. 2021). We also compare to deep-learning free classical SPSR (Kazdan et al. 2013).
>
> In order to enrich the comparison further and give the reader even more perspective, we include the following additional comparisons in the updated version of the paper, and show the updated respective tables here bellow:
>
> - We add a comparison to recent unsupervised method GridPull (Chen et al. 2023) in the shapenet comparison in Table 1.
> - We add a comparison to recent unsupervised method GridPull (Chen et al. 2023) in the Faust comparison in Table 2.
> - We add a comparison to generalizable method POCO (Boutch et al. 2022) in the Faust comparison in Table 2.
> - We add a comparison to generalizable method CONet (Peng et al. 2020) in the Faust comparison in Table 2.
> - We add a comparison to the sota generalizable method NKSR (Huang et al. 2023) in the Faust comparison in Table 2.
> - We add a comparison to a generalizable method that combines priors and test-time optimization, and which also addresses specifically sparse input: On-Surf. Priors (Ma et al. 2022) in the Faust comparison in Table 2.
> - We add a numerical comparison to methods DIGS and NTPS in Table 5, the sparse reconstruction experiment in benchmark SRB.
>
> Hence we present here the updated versions of tables 1, 2 and 5 of the paper. We note that the comparison to generalizable methods outlines the fact that priors learned on shapenet can fail to generalize for OOD samples or samples with different size from the training corpus.
>
>
> |            | CD1   | CD2   | NC    | FS    |
> |------------|-------|-------|-------|-------|
> | POCO       | 0.308 | 0.002 | 0.934 | 0.981 |
> | Conv-onet  | 1.260 | 0.048 | 0.829 | 0.599 |
> | On-Surf    | 0.584 | 0.012 | 0.936 | 0.915 |
> | NKSR       | 0.274 | 0.002 | 0.945 | 0.981 |
> | SPSR       | 0.751 | 0.028 | 0.871 | 0.839 |
> | Grid-Pull  | 0.495 | 0.005 | 0.887 | 0.945 |
> | NTPS       | 0.737 | 0.015 | 0.943 | 0.844 |
> | NP         | 0.269 | 0.003 | 0.951 | 0.973 |
> | Ours       | **0.220** | **0.001** | **0.956** | **0.981** |
> Table1: Faust  reconstructions from sparse noisy unoriented point clouds.
>
> |             | CD1  | CD2 | N| FS |
> |------------|----------|---------|--------|--------|
> | SPSR       | 2.34     | 0.224   | 0.74   | 0.50   |
> | OG-INR     | 1.36     | 0.051   | 0.55   | 0.55  |
> | NP         | 1.16     | 0.074   | 0.84   | 0.75  |
>  | Grid-Pull  | 1.07     | 0.032   | 0.70   | 0.74  |
> | NTPS       | 1.11     | 0.067   | 0.88   | 0.74  |
>  | Ours       |**0.76** | **0.020**   | **0.88**   | **0.83**   |
> Table 2: Shapenet  reconstructions from sparse noisy unoriented point clouds.
>
> |        | Sparse   |          | Dense |  |
> |--------|---------------|-------------|-------|-------|
> |        | CD            | HD          | CD    | HD    |
> | SPSR   | 2.27          | 21.1        | 1.25  | 22.59 |
> | DiGs   | 0.68          | **6.05** |**0.19**  | **3.52**  |
> | OG-INR | 0.85          | 7.10        | 0.20  | 4.06  |
> | NTPS   | 0.73          | 7.78        | -     | -     |
> | N-Pull | 0.58          | 8.90        | 0.23  | 4.46  |
> | Ours   | **0.49** | 8.17        | **0.19**  | 3.84  |
> Table 5: SRB  reconstructions from sparse noisy unoriented point clouds.

---

> ### Author Response · Authors · 2023-11-20
>
> ### Lastly, it is generally believed that the performance on the original samples tends to decrease after adversarial training. It would be helpful to provide additional explanations as to why this performance decrease is reasonable.
>
> We thank the review for raising this concern and we clarify our positioning bellow.
> In the adversarial training literature, such a trade-off has been observed empirically in different datasets [1], [2].  This has led prior work to claim that this tradeoff  may be inevitable for many classification tasks [3], [4]. However, many recent papers challenged this claim. [5] showed theoretically that  this tradeoff is not inherent. Rather, it is a consequence of current robustness methods. These findings are corroborated empirically in many recent work [6], [7], [8].
>
> However, while we formulate our problem as a (Distributional) robust optimization our setting differs from the aforementioned work in many aspects. First, it is not a classification setting. Many theoretic arguments in [3] and [6] are based on whether adversarial samples remain at the same class manifold as their natural counterpart. Additionally, the drop in accuracy is explained in [7] by the distribution mismatch between natural and adversarial images. Such a discrepancy is not applicable to our setting as the network should perform consistently in the 3D space including “spatial adversaries”. Furthermore, our baseline relies on a pseudo-labeling strategy (i.e. associating a query point to its nearest input point) that introduces noise as the input gets sparser. Hence, Our approach is meant to robustify the learning process against this noise, which provides two main additional benefits:
>
> - **Regularization**:  We observed that when the network starts overfitting it creates blobs and bumps on the surface of the shape in addition to hallucinations. This means that some query points are pulled far away from their associated input points. While the loss in theses points can be high, the average loss is dominated by “easy samples” at advanced stages of the training. For the same reason the network tend to ignore some parts of the shape where the sampling density is very low (like body extremities in Faust in Figure 4, and the floor in the Scene copy room of the [uploaded video](https://anonymous.4open.science/r/Adv-F74C/) ). Our approach regularizes the network by “flattening” the loss landscape such that the loss behavior on the “training” query points is generalized to “spatial adversaries”.  We note that the link between robust optimization and the curvature of the loss have been previously observed (empirically) in [9].
>
> - **Informative samples**: At advanced training stages, our training strategy provides the network with query points that helps in refining the shape function as they carry more information for the network. In this regard, we would like to stress when it comes to spatial adversaries that the  consistency of the quey-pseudo-label association is crucial to the success of our method. This is possible thanks to locally adaptive perturbation radii. As shown in ablation, a global radius can result in over-regularization. We note that a similar observation, i.e. locally adaptive perturbations breaks/improves the robustness-accuracy trade-off, have been observed in [6], [10], [11].

---

> > ### Author Response · Authors · 2023-11-20
> >
> > We will add this analysis to the next version for to provide the reader with a more complete insight on our work.
> >
> > [1] Aditi Raghunathan, Sang Michael Xie, Fanny Yang, John C duchi,and Percy Liang. Adversarial training can hurt generalization. arXiv preprint arXiv:1906.06032, 2019.
> >
> > [2] Aleksander Madry, Aleksandar Makelov, Ludwig Schmidt, Dimitris Tsipras, and Adrian Vladu. Towards deep learning models resistant to adversarial attacks. In International Conference on Learning Representations, 2018.
> >
> > [3] Dimitris Tsipras, Shibani Santurkar, Logan Engstrom, Alexander Turner, and Aleksander Madry. Robustness may be at odds with accuracy. arXiv.org, abs/1805.12152, 2018.
> >
> > [4]  HongyangZhang,YaodongYu,JiantaoJiao,EricXing,Lau- rent El Ghaoui, and Michael Jordan. Theoretically princi- pled trade-off between robustness and accuracy.
> >
> > [5] Yao-Yuan Yang, Cyrus Rashtchian, Hongyang Zhang, Rus- lan Salakhutdinov, and Kamalika Chaudhuri. A closer look at accuracy vs. robustness. In Advances in Neural Information Processing Systems (NeurIPS), 2020.
> >
> > [6] David Stutz, Matthias Hein, and Bernt Schiele. Disentan- gling adversarial robustness and generalization. In IEEE Conference on Computer Vision and Pattern Recognition (CVPR), 2019.
> >
> > [7] Cihang Xie, Mingxing Tan, Boqing Gong, Jiang Wang, Alan L Yuille, and Quoc V Le. Adversarial examples im- prove image recognition. In Proceedings of the IEEE/CVF International Conference on Computer Vision (ICCV), pages 819–828, 2020.
> >
> > [8] Charles Herrmann, Kyle Sargent, Lu Jiang, Ramin Zabih, Huiwen Chang, Ce Liu, Dilip Krishnan, and Deqing Sun. Pyramid adversarial training improves vit performance. arXiv preprint arXiv:2111.15121, 2021.
> >
> > [9] Seyed-Mohsen Moosavi-Dezfooli, Alhussein Fawzi, Jonathan Uesato, and Pascal Frossard. Robustness via curvature regularization, and vice versa. In CVPR, 2019.
> >
> > [10] Yogesh Balaji, Tom Goldstein, and Judy Hoffman. Instance adaptive adversarial training: Improved accuracy trade- offs in neural nets. arXiv preprint arXiv:1910.08051, 2019.
> >
> > [11] Jingfeng Zhang, Jianing Zhu, Gang Niu, Bo Han, Masashi Sugiyama, and Mohan Kankanhalli. Geometry-aware instance-reweighted adversarial training. In International Conference on Learning Representations (ICLR), 2021a.

---

> ### Author Response · Authors · 2023-11-22
>
> Thank you very much for your reply.
>
> 1. From the perspective of learning an SDF through “Neural pulling”, the “sparse noise and point displacement noise” can be conceptually very similar and can lead to the same outcome:
>
> - If the input point cloud is sparse, queries are pulled to “badly matched” input points. These noisy pseudo-labels are in this case the GT perfect labels displaced with some offset i.e. noise.
>
> - If the input point cloud is noisy, queries are pulled to these “inaccurate” input points. These noisy pseudo-labels are in this case the GT perfect labels displaced with some offset i.e. noise.
>
> Our key observation is that the learning process of our baseline is prone to overfitting in the sparse setting. When the input is additionally noisy, this exacerbates the overfitting effect on the learned implicit representation. In both cases, a noise due to the displacement of the GT perfect labels is introduced in the supervision of the query points, which can affect both the SDF function and gradient orientation. This is true whether this displacement keeps the points on the surface of the shape (sparsity) or can take them outside the surface (input noise). As it can be seen in Gargoyle [example](https://anonymous.4open.science/r/Adv-F74C/srb/gargoyle/NP.mp4 )  from the SRB Benchmark the baseline first produces a very smooth shape and when it tries to refine it, it ends up overfitting the noise introduced in the supervision. At this stage, further fitting on easy samples (predominant samples) means overfitting on this noise. The samples that can benefit the implicit representation can be drowned within easy samples. Our adversarial training strategy, instead focuses on samples that can still benefit the network, which prevents the aforementioned overfitting while refining the implicit representation.

---

> ### Author Response · Authors · 2023-11-22
>
> 3. We are sorry if we misunderstood your initial comment. Indeed we agree that classifying the methods we compare to in a more intuitive manner would help the reader. We will clarify in our results section that methods we compare to can be classified as shown bellow.
>
>     • Unsupervised methods
>
>         ◦ Per shape neural fitting
>             ▪ Can handle sparse PC density
>                      NTPS (Chen et al. 2023)
>                      NSpline (Williams et al. 2021) [Requires normals]
>             ▪ Generic PC density
>                 • Fully implicit
>                     NP (Ma et al. 2021) [baseline]
>                     DIGS (Ben-Shabat et al. 2022)
>                     SAP (Peng et al. 2021)
>                     PHASE (Lipman 2021)
>                     NDrop (Boulch et al. 2021)
>                     IGR (Gropp et al. 2020)
>                     SIREN (Sitzmann et al. 2020)
>                     SAL (Atzmon & Lipman 2020)
>                 • implicit + grid
>                     OG-INR (Koneputugodage et al. 2023)
>                     GridPull (Chen et al. 2023)
>         ◦ Optimization (learning-free)
>                     SPSR (Kazdan et al. 2013)
>
>     • Supervised method
>
>         ◦ Neural feed-froward prediction
>             ▪ Can handle sparse PC density
>                     POCO (Boutch et al. 2022)
>                     CONet (Peng et al. 2020)
>                     NKSR (Huang et al. 2023)
>         ◦ Neural feed-forward prediction + Per-shape neural fitting
>             ▪ Can handle sparse density
>                     On-Surf. Priors (Ma et al. 2022)
>
> We believe that our comparisons covers a variety of sota and classical methods, showcasing various priors. We note that the methods present in the survey suggested by the reviewer can fit within our classification as follows (differently, we compare to more recent variants in most cases, and we compare to our most direct competitor: NTPS)
>  • Unsupervised methods
>
>         ◦ Per shape neural fitting
>             ▪ Can handle sparse PC density
>
>             ▪ Generic PC density
>                 • fully implicit
>                      SALD
>                      IGR
>                 • implicit + grid
>                      LIG
>         ◦ Optimization (learning-free)
>                      SPSR
>                      GD
>                      BPA
>                      RIMLS
>
>    • Supervised methods
>
>         ◦ Neural feed-froward prediction
>             ▪ Can handle sparse PC density
>                     OccNet
>                     IMLSNet
>             ▪ Generic PC density
>                     Points2Surf
>                     DSE
>                     ParSeNet
>         ◦ Neural feed-forward prediction + Per-shape neural fitting
>             ▪ Can handle sparse density
>                    DeepSDF

---

> > ### Comment · Reviewer_akZJ · 2023-11-23
> >
> > Thank you for the response. Most of my concerns have been addressed, so I will raise the score to 6: marginally above the acceptance threshold.

---

### Official Review · Reviewer_PU6C · 2023-11-08

**Soundness:** 3 good
**Presentation:** 3 good
**Contribution:** 3 good
**Rating:** 6
**Confidence:** 3

**Summary:**

The paper considers the task of learning neural SDFs from sparse point cloud data. The typical approach for learning such SDFs without external supervision is to sample  **query** points around each point in the pointcloud, and optimizing a loss that pulls the query points toward the closest point on the neural surface (Neural-Pull, Ma et al 2021). The authors observe that such methods are prone to overfitting (the training loss continues to drop when the validation chamfer error grows). The authors propose a simple solution to this problem. After sampling query points near the input point clouds, one can find additional **adversarial** points near the query points that maximize the SDF loss. The authors demonstrate empirically that training with these additional adversarial points improves the quality of SDFs recovered from point clouds on a range of 3D datasets.

**Strengths:**

* The paper is clear to read and the content is easy to follow.
* The related work is thorough and clearly positions prior work.
* The proposed method is simple and effective.
* The results demonstrate state-of-the-art performance compared to existing methods.
* The approach is well-motivated from the perspective of robustness and hard sample mining for improved generalization

**Weaknesses:**

* Insufficient analysis of how overfitting is manifested and how it is addressed: The central motivation for the approach is that NeuralPull and related works experience overfitting, but it’s not clear what it looks like to experience overfitting. For instance, maybe the surface starts smooth and then becomes bumpy around the input points? It would be illuminating to see some actual NeuralPull reconstructions at different levels of overfitting to be clear about what problem the proposed work is addressing.
* The task is unrealistic: For this method (and related approaches) to work, the point clouds must be roughly uniform in coverage over the object. At a minimum, there must be good coverage of points across most of the object (and this is why the method struggles on thin structures as shown in Figure 3). This is reflected in the evaluation setup, where all datasets start with a high-quality mesh which is then subsampled with uniform density into a point cloud, which is then used to reconstruct a surface.  However, in real-world point clouds, the sampling will never be uniform in density or coverage. For instance, in point clouds captured via depth cameras (one of the most common ways to acquire a point cloud), most of the points will be biased toward regions of high-frequency texture with very few points on textureless surfaces.
* (minor) The differences in Figure 5 are difficult to observe even with the red box. Figure 1 text is too small.

**Questions:**

* Why is the reconstruction quality better than NeuralPull? From Figure 1, it seems that the primary benefit of the red line over the baseline is that there is less overfitting. However, the red line does not seem to go significantly lower than the green line for the Chamfer validation. In other words, I would think that the primary benefit of the proposed work is in robustness/less hyperparameter tuning and not outright reconstruction quality.
* Related to the above, how were hyperparameters for prior works chosen, namely for which epoch to stop at? As the paper notes, which epoch to stop at has an immense effect for NeuralPull. Perhaps the fairest comparison would be to consider 3 setups for each baseline: stop at the optimum iteration based on test error, at a fixed early-ish iteration, and at a fixed very late iteration. For the proposed approach, I would expect all 3 to give similar performance whereas the gap would be large for the baselines. This would also help demonstrate the robustness of the proposed work.

---

> ### Author Response · Authors · 2023-11-20
>
> We would like to thank the reviewer for their insightful comments and constructive criticism of our work. We address below the review. We note that we will upload a newer version of the paper PDF shortly, which will contain new figures and updates on tables referred to in the rebuttal.
>
> ###  Insufficient analysis of how overfitting is manifested and how it is addressed.
>
> We thank the reviewer for raising this point.
>
> Firstly, as  shown in **Figure 1**  in the paper (and in all numerical comparison tables), the overfitting is manifested numerically in the increase of the main comprehensive metric of reconstruction, i.e. Chamfer distance to the ground-truth.
>
> Following the request of the reviewer, we will add a figure in the paper showing qualitative examples of this overfitting for our baseline NP (Ma et al. 2021) at different iterations. We also provide hereby **a link to videos [ link to videos](https://anonymous.4open.science/r/Adv-F74C) showing this overfitting in examples from the SRB and 3D Scenes Benchmarks**, to showcase both object and scene level overfitting. We show both our method and the baseline’s behavior.
>
> In the case where the baseline did not entirely diverge from the very beginning, as seen in these videos, we observe the validation chamfer gap (shown in Figure 1 in the paper) is synonymous to qualitative deterioration in the extracted shape with symptoms varying between shapes. Specifically, we mainly observe the following visual phenomena:
> - **Shape hallucinations**, i.e. shape appearing in places erroneously, where there should be no shape.
> - **Shape missing** in places where there should be shape.
> - Shape becoming progressively wavy, **bumpy** and noisy, until it starts breaking into different components in many cases. In extreme cases we can end up with separate **isolated components** around the input points, or clusters of input points.
>
> This visual analysis underpins the danger of relying solely on easy samples for learning from a sparse and noisy input point cloud, especially without normal guidance. The shape bumps and hallucinations  mean that some query points will be incorrectly projected (with the NP gradient based projection mechanism). However, although the loss at these points can be high, we believe it can still be **dominated in average by the loss at easy samples**. For the same reasons, missing shape means some **parts of the shape can be ignored by the model** as it can be shown in the Faust comparative reconstruction (Figure 4), where body extremities are overlooked by the baseline, or the missing floor in scene copy room in the 3D scene benchmark video. Supervising the model with the **worst-case samples** not only acts as a **regularization preventing the model from overfitting** on the query points, but also provides the model with more informative samples that can help **refine its implicit representation** of the shape.
>
> We will include a thorough visual analysis of overfitting, in addition to a clearer explanation of the reasons of our improved performance accordingly in the next version of the paper.
>
> ### The task is unrealistic
>
> We understand the reviewer’s general concern about this point. However we merely work here within the **conventional well-established experimental literature setups** used across many current competing sota of reconstruction from point cloud (e.g. DIGS (Ben-Shabat et al. 2022), NP (Ma et al. 2021), OG-INR (Koneputugodage et al. 2023), SAP (Peng et al. 2021), IGR (Gropp et al. 2020), SIREN (Sitzmann et al. 2020), (SAL (Atzmon & Lipman 2020), PHASE (Lipman 2021), NSpline (Williams et al. 2021), … etc), which serves as a benchmarking tools for ideas, admittedly with varying levels of realism.
>
>   We would like to note additionally that reconstruction from **non-partial point sets** is a classical problem with various downstream applications in graphics and vision. Traditionally, it is approached through combinatorical methods (e.g. Alpha shapes, Voronoi diagram, Voronoi triangulation etc) or learning-free classical optimization (e.g. Poisson, APSS, MLS, etc). More recently, deep learning, most notably through implicit representations, triumphed in this problem setting either in the supervised (CONet (Peng et al. 202O), SAP (Peng et al. 2021), POCO (Boutch et al. 2022), … etc) or unsupervised settings (IGR (Gropp et al. 2020), NP (Ma et al. 2021), DIGS (Ben-Shabat et al. 2022), … etc).
>
> While full scans can be aggregated from different views, we agree that there are real life scenarios that require reconstruction from only partial point cloud and/or non uniform clouds. We focus in this work on a different set of challenges, i.e. **noise and sparsity of inputs**, and we contribute an **original method** that addresses the difficulties that arise from these challenges. As pointed out by most reviewers, we believe our problem formulation and results to be relevant and of considerable value to the community in this context.

---

> ### Author Response · Authors · 2023-11-20
>
> ###  The differences in Figure 5 are difficult to observe even with the red box. Figure 1 text is too small
> Thank you for noticing this. We will try to clarify the differences in Figure 5 with qualitative zooms and additional commentary in the next version of the paper, as well as increasing the font size in Figure 1.
>
> ### Why is the reconstruction quality better than NeuralPull? From Figure 1, it seems that the primary benefit of the red line over the baseline is that there is less overfitting. However, the red line does not seem to go significantly lower than the green line for the Chamfer validation. In other words, I would think that the primary benefit of the proposed work is in robustness/less hyperparameter tuning and not outright reconstruction quality.
>
> We kindly beg to differ here with the reviewer. Actually, if we observe the reconstruction from sparse data curves (dashed lines) in Figure 1, the difference between our best model and the best baseline model is significant. Perhaps the fact that we show Chamfer values for unit box normalized meshes is a bit confusing in this case, and we will clarify it in the caption in the next version.
>
> In fact, this very gap between those plots amounts to **L1 chamfer** (10^-2) dropping in average  from 1.16  to 0.76 in the ShapeNet benchmark (Table 1) for instance. Our improvements over the baseline across all the benchmarks shown in the paper are considered as healthy and **important improvements** in the community, as can be witnesses in comparative tables in other literature.
>
> Regarding the second part of the question, we would like to clarify again (as in our previous answer) that supervising the model with the worst-case samples not only acts as a **regularization** preventing the model from overfitting on the query points, but also provides the model with more **informative samples**. This can help refine the implicit prediction and hence **improve reconstruction performance**.
>
> In fact, although the loss at the overfitting queries will be high, we believe it can still be overshadowed and drowned in average in the loss of easy queries. For the same reasons, some parts of the shape can be ignored by the model as it can be shown in the Faust comparative reconstructions (Figure 4), where body extremities in particular are neglected by the baseline. Our contribution remedies this behavior, which leads to both **reduced overfitting and improved performance**. We note that **these positive outcomes** are not incompatible. Besides, our superior **results** to both competition and baseline across all benchmarks **demonstrate this conclusion empirically**.
>
> ### Related to the above, how were hyperparameters for prior works chosen, namely for which epoch to stop at? As the paper notes, which epoch to stop at has an immense effect for NeuralPull. Perhaps the fairest comparison would be to consider 3 setups for each baseline: stop at the optimum iteration based on test error, at a fixed early-ish iteration, and at a fixed very late iteration. For the proposed approach, I would expect all 3 to give similar performance whereas the gap would be large for the baselines. This would also help demonstrate the robustness of the proposed work.
>
>  We believe it is not realistic to use the ground-truth to stop the training as such information is naturally unavailable in real situations. Hence, in the interest of **practicality and fairness** in our comparisons, we decide the evaluation epoch for **all the methods** for which we generated results (including our main baseline)**in the same fair way**: we chose the best epoch for all methods alike in terms of chamfer distance between the reconstruction and the input point cloud. We remind that this is **the only possible way to pick the evaluation model in real life**. We will clarify this point in the next version of the manuscript.
> As pointed out in our answer to the previous question, our approach does not improve only robustness, but boosts performance as well. Under the same best epoch selection procedure (Minimal chamfer distance between reconstruction and input), our method outperforms its baseline and the competition.

---

> > ### Comment · Reviewer_PU6C · 2023-11-21
> >
> > I thank the authors for the detailed response and clarifications. In particular, I appreciated the qualitative video showing the optimization progress comparing NeuralPull with the proposed approach, in which it is clear that NeuralPull exhibits overfitting.
> >
> > However, this still relates to my original concern that if training loss evaluated via chamfer distance between the input point cloud and the reconstruction is used to determine the stopping point for NeuralPull (and other baselines), then the reconstructions produced by the baselines are necessarily going to be overfit and low-quality. The loss curve from Figure 1 suggests that NeuralPull's training loss decreases monotonically, so if the evaluation is done at a late epoch, then the reconstructions will be severely overfit.
> >
> > I agree with the authors that the evaluation I suggested is not practical or fair, but that's missing the point. Such a baseline is meant to be an "oracle" comparing the best-case reconstruction of NeuralPull with the proposed approach. Such an oracle could also be applied to the proposed approach to fairly compare the "best-case" reconstruction of both methods. In the current evaluation, it is essentially comparing a "worst-case" (over-fit) reconstruction of NeuralPull with the "average-case" reconstruction of the proposed approach (please correct me if I am misunderstanding).

---

> ### Author Response · Authors · 2023-11-21
>
> Thank you very much for your answer.
>
> Our **training loss is not the metric that we use to pick the validation model**, neither for us nor the baselines.
>
> We apologize if our presentation of the loss was not clear enough, but **the training loss is not “evaluated via chamfer distance between the input point cloud and the reconstruction”**. The training loss is the distance between input points and nearby query points projected via the SDF function to its zero level set. The differences are:
>
> (1) This **loss only loosely approximates one side part of the two-way chamfer**, also referred to as **accuracy** in literature. The other part of the chamfer metric is important, often referred to as **completeness**. Combining **both parts is crucial for chamfer to approximate a proper Wasserstein distance (earth mover distance)**.
>
> (2) **This approximation of one part of the two-way chamfer is not accurate**, as the projected query points **are not necessarily uniformly sampled from the zero level set**. When we compute chamfer from reconstructions, **the samples must be uniformly generated from the explicit mesh**, i.e. the zero level set.
>
> To illustrate our point and complete Figure 1  in the paper (in the sparse case), we show a [plot](https://anonymous.4open.science/r/Adv-F74C/ShapeNet/plots/cd1_input.png) of the chamfer distance between reconstruction and input, and we join also the corresponding plot in the paper that was showing chamfer distance to the ground-truth. Notice that the curves behave overall similarly, and hence either would lead to the same selection criterion. Thusly, **using Chamfer distance to input leads to the same results as using chamfer distance to GT**.
>
> To prove our point further we show here **numerical evidence**:
>
> we recompute the evaluation of the ShapeNet experiment in Table 1 of the paper. Here, models are picked with the GT chamfer. Our performance gap with respect to the baseline is evidently the same as before, as **numbers do not change much overall**.
>
> |       | With GT CD1 | With Input CD1 |
> |-------|-------------|----------------|
> | Lamp  | 1.51        | 1.52           |
> | Table | 1.05        | 1.07           |
> | Chair | 0.89        | 0.91           |
> | Mean  | **1.15**        | **1.16**          |
>  Table: Baseline results using CD1 to the  ground truth and to the input point cloud to pick the validation model.
>
> We remind that the **full two-way chamfer with uniform sampling** between reconstruction mesh and input is not only the best, but also the **only** metric we have at our disposal to decide convergence.

---

> > ### Comment · Reviewer_PU6C · 2023-11-22
> >
> > Oh got it, I understand now. Thank you for the clarification. I am not sure why that was not clear before.
> >
> > I have updated my review. I would consider my concerns largely addressed and would encourage the authors to update the pdf with the new changes.

---

### Author Response · Authors · 2023-11-23

We would like to thank the reviewers for their valuable feedback. Based on the discussion with the reviewers, we summarize bellow the main changes in our revised version of the Paper and the supplementary material:

### PU6C

- we include a visual analysis of overfitting in Introduction (shape hallucinations, shape missing, bumps)
- for lack of space in the paper, we join the video showing visual examples of overfitting in the SRB and 3D Scenes Benchmarks in the suplemantary material.
- we include a clearer explanation of the reasons for our improved performance in introduction (Regularization+informative samples)
- we add a zoom effects for better visualization in Figure 5
- we increase the size of the text in Figure 1
- we precise “Chamfer values for unit box normalized meshes” in caption Figure 1
- we clarify that all eval models are picked with chamfer to input in Results.

### AkZJ

- we clarify why our strategy is effective with sparsity and not just noise in introduction (both sparsity and noise result in noisy peudo-labels for us)
- In Table 1 (Shapenet benchmark), we add method GridPull (Chen et al. 2023)
- In Table 2 (Faust benchmark), we add methods GridPull (Chen et al. 2023), NKSR (Huang et al. 2023), POCO (Boutch et al. 2022), CONet (Peng et al. 2020), On-Surf. Priors (Ma et al. 2022).
- In Table 5 (Sparse SRB benchmark), we add methods DIGS (Ben-Shabat et al. 2022) and NTPS (Chen et al. 2023).
- we clarify our positioning wrt adversarial training literature in Related work
- we provide additonal elements clarifying the benefits of our contritution (Regularization+informative samples) in introduction
- We provide a clearer classification of our competition in Results

### IVm1

- we add discussion about NC performance in Limitations (excess regularization)
- we add a comparison between our adversarial samples and just more normal samples in Ablations
- we add a comparison between our adversarial learning and baseline with increasing size of input in Ablations

### ZZdy

- we add the full derivation Eq 6-8 in method.
- we add text in Results and videos in supplemantary material justifying why we do not use extremely sparse inputs
- we bolded best values in Table 3
- we clarified the metric definition for Tables 4 and 5
- In Table 5 (Sparse SRB benchmark), we add methods DIGS (Ben-Shabat et al. 2022) and NTPS (Chen et al. 2023).
- we add a comparison between our adversarial samples and just more normal samples in Ablations
- we clarified global radius definition in Table 6
- we add a comparison between the hybrid loss and adversarial loss alone in Ablations